# Relaxing Local Robustness

**Klas Leino**
Carnegie Mellon University
kleino@cs.cmu.edu

**Matt Fredrikson**
Carnegie Mellon University
mfredrik@cs.cmu.edu

## Abstract

Certifiable *local robustness*, which rigorously precludes small-norm *adversarial examples*, has received significant attention as a means of addressing security concerns in deep learning. However, for some classification problems, local robustness is not a natural objective, even in the presence of adversaries; for example, if an image contains two classes of subjects, the correct label for the image may be considered arbitrary between the two, and thus enforcing strict separation between them is unnecessary. In this work, we introduce two relaxed safety properties for classifiers that address this observation: (1) *relaxed top-k robustness*, which serves as the analogue of top-k accuracy; and (2) *affinity robustness*, which specifies which sets of labels must be separated by a robustness margin, and which can be $\epsilon$-close in $\ell_p$ space. We show how to construct models that can be efficiently certified against each relaxed robustness property, and trained with very little overhead relative to standard gradient descent. Finally, we demonstrate experimentally that these relaxed variants of robustness are well-suited to several significant classification problems, leading to lower rejection rates and higher certified accuracies than can be obtained when certifying "standard" local robustness[1].

## 1 Introduction

The discovery of *adversarial examples* [9, 24, 28] has led to security concerns in deep learning. A growing body of work has sought to address this problem by providing provable guarantees that a model's predictions are robust to small-norm perturbations [3, 5, 8, 13, 16, 21, 22, 25, 29, 31, 32, 33]. This objective is typically captured by ensuring that a model satisfies point-wise *local robustness*; i.e., given a point, $x$, the model's predictions must remain invariant over the $\epsilon$-ball centered at $x$. Local robustness is accompanied by a metric, *verified-robust accuracy* (VRA), which corresponds to the fraction of points that are both correctly classified and locally robust.

In some contexts, however, it is not always clear that VRA is the most desirable objective or the most natural metric for measuring a model's success against adversaries. For example, in some contexts, *not all adversarial examples are equally bad*—this may reflect simply that a mistake is understandable, even if it was caused by an adversary (e.g., if a model mistakenly predicts an image of a leopard to be a jaguar); or that the correct label may be arbitrary in certain cases (e.g., if an image contains two classes of subjects).

For similar sorts of reasons, some computer vision tasks often use *top-$k$ accuracy* as a benchmark metric that relaxes standard accuracy, allowing a prediction to be considered correct so long as the correct label appears among the model's $k$ highest logit outputs. In many applications, top-$k$ accuracy is considered a more suitable metric/objective than standard top-1 accuracy [1]; meanwhile, studied robustness properties are typically only defined with respect to a *single* predicted class. Thus, as part of this work, we introduce an analogous relaxation of local robustness to top-$k$ accuracy, which we call *relaxed top-$K$ (RTK) robustness* (Section 3, Definition 3). Moreover, we demonstrate how

---

[1]our code is publicly available at https://github.com/klasleino/gloro

a neural network can be instrumented in order to naturally incorporate certifiable RTK robustness into its learning objective, making runtime certification of this property essentially free (Section 5).

In addition to applying to only top-1 predictions, standard robustness also focuses specifically on the problem of *undirected* adversarial examples. More concretely, adversarial examples are obtained broadly via two categories of attacks: (1) *evasion* (undirected) attacks, and (2) *targeted* (directed) attacks. Local robustness provides guarantees against the former, while the latter, which requires only a weaker guarantee, has remained largely unexplored by work on certifiable robustness. In this work we introduce *affinity robustness* (Section 4, Definition 4), which captures resistance to specified sets of directed adversarial attacks. We show that certifiable affinity robustness can also be achieved in a manner similar to RTK robustness (Section 5).

In recent work, Leino et al. [22] introduced a concept of $\epsilon$-*global-robustness*, which can be thought of as requiring regions of the input space that are labeled as different classes to be separated by a margin of $\epsilon$, with any interstitial space labeled as $\perp$, signifying rejection. By relaxing the notion of robustness, we essentially allow certain classes to be grouped together without any intervening rejected space. This gives rise to a spatially-arranged hierarchy of classes that is conveyed through the robustness guarantee (see Figure 2 in Section 3 for an example). Interestingly, we find that models trained with the objective of RTK robustness often form a logical hierarchy even without supervision. Additionally, affinity robustness provides a way to supervise the hierarchy that forms, giving finer control over the spatial relationships between the various labels in the model's decision surface.

In summary, the primary contributions in this work are as follows:

- We introduce *RTK robustness*, a notion of model robustness compatible with top-$k$ accuracy, and *affinity robustness*, a notion of model robustness that captures resistance to particular sets of *directed* adversarial attacks.

- We show how to construct models that can be *efficiently certified* against these two properties.

- We show that applying these relaxed robustness variants to suitable domains leads to certifiable models with lower rejection rates and higher certified accuracy.

- We show that these relaxed robustness variants lead to interesting properties regarding how a network's prediction space is laid out, and that this layout of classes can be supervised to impart *a priori* hierarchies.

## 2   Related Work

Robustness certification for deep networks has become a well-studied topic, and numerous certification approaches have been proposed [3, 5, 8, 13, 16, 21, 22, 25, 29, 31, 32, 33]. However, virtually all of this work has focused specifically on certifying guarantees against $\ell_p$-norm-bounded, *undirected* attacks on a model's *top* prediction. By contrast, we introduce two relaxed variants of this standard robustness property that have not formerly been studied, which we subsequently show how to certify. Although our proposed definitions are agnostic to both the chosen norm and the method used to certify them, we focus on *deterministic* robustness certification (as opposed to *stochastic* certification, e.g., Randomized Smoothing [3, 20]) for the $\ell_2$ norm. We build our certification approach from a recently proposed method by Leino et al. [22], which has been demonstrated to be among the most scalable state-of-the-art methods for deterministic $\ell_2$ certification.

Much of this work focuses on generalizing the most commonly-used threat model characterizing what it means to be "robust", namely, $\ell_p$ local robustness, which stipulates prediction invariance over perturbations within a small $\ell_p$ ball. Other work has also considered generalizations of this robustness definition, though not typically in the context of certification. E.g., previous literature has proposed invariance under unions of multiple $\ell_p$ balls [4, 30], invariance to rotations and translations [6], and invariance to inconspicuous, physically-realizable perturbations [2, 7, 18, 27].

The first of the two robustness definitions we propose, RTK robustness, is closely related to top-$k$ accuracy. Recently, Jia et al. [12] also proposed a definition that aims to capture certified robustness for top-$k$ predictions. However, their work differs from ours in two key ways. First, their certification method is derived from Randomized Smoothing, which gives a stochastic guarantee and relies on hundreds of thousands of samples for conclusive certification. More importantly, Jia et al. evaluate their property with respect to the *ground truth* class of the point in question, making it unclear how

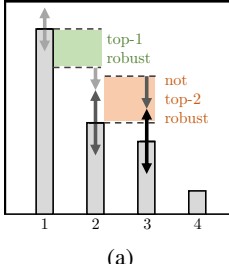 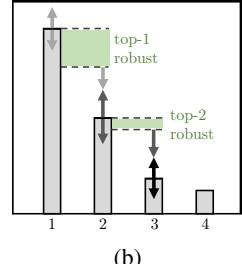

(a)                    (b)

Figure 1: **(a)** Example of network outputs that demonstrate top-$k$ robustness is not a relaxation of standard local robustness. The gray arrows denote bounds on the amount each logit can change within a radius of $\epsilon$. We see that these bounds are not sufficient for class 2 to surpass class 1; however, the bounds are sufficient for class 3 to surpass class 2. Therefore, the point in this example is top-1 robust, but *not* top-2 robust. **(b)** Example of network outputs that are simultaneously top-1 robust and top-2 robust.

one might certify this property in such a manner on unlabeled points, e.g., those seen by the model in deployment. We provide more discussion on this problem and how our work addresses it in Appendix A in the supplementary material. As we will see, devising a robustness analogue to top-$k$ accuracy in a manner that relaxes standard robustness and does not depend on the ground truth is a subtle task—we address this issue carefully in Section 3.

Our second proposed robustness definition, affinity robustness, is related to targeted adversarial attacks [28], which have only rarely been studied in the context of certifiable defenses [10].

## 3 Relaxed Top-K Robustness

In this work, we consider relaxations of the standard notion of local robustness (Definition 1) that may serve as a better learning objective and evaluation metric in some contexts, e.g., those where not all adversarial examples are equally bad.

**Definition 1 (Local Robustness)** *A model, $F$, is $\epsilon$-locally-robust at point, $x$, w.r.t. norm, $||\cdot||$, if*

$$\forall x' \; . \; ||x - x'|| \leq \epsilon \implies F(x) = F(x')$$

We begin in this section by introducing a notion of robustness that is inspired by the relaxed accuracy metric, top-$k$ accuracy. Top-$k$ accuracy is a common benchmark metric in vision applications such as Imagenet, where the class labels are particularly fine-grained, and may even be arbitrary on some instances. Furthermore, top-$k$ accuracy has been studied as a learning objective in its own right [1], and has been identified as desirable in the context of robustness certification [12].

In order to create a notion of relaxed robustness that is analogous to top-$k$ accuracy, we will consider the network to output a *set* of classes rather than a single class. Let us define the following notation representing the set of the top $k$ outputs of a model: let $f$ be the function computing the logit values of a neural network, and let $f^k(x)$ be the $k^{\text{th}}$-highest logit output of $f$ on $x$. We then define $F^k(x) = \{j : f_j(x) \geq f^k(x)\}$, that is, $F^k$ is the set of classes corresponding to the top $k$ outputs of $f$. Using this notation, we define *top-$k$ robustness* (Definition 2), which requires that the set of classes with the $k$ highest logits remains invariant over small-norm perturbations. We note that if we had a single class of interest, $c$, e.g., the ground truth, we could simply require that $c$ remain in $F^k$ under small perturbations [12]; however, top-$k$ accuracy is useful precisely because *any* of the classes in $F^k$ could be correct, meaning that all classes in $F^k$ should be treated equally and guarded against perturbations.

**Definition 2 (Top-k Robustness)** *A model, $F$, is top-$k$ $\epsilon$-locally-robust at point, $x$, w.r.t. norm, $||\cdot||$, if*

$$\forall x' \; . \; ||x - x'|| \leq \epsilon \implies F^k(x) = F^k(x')$$

While top-$k$ robustness may appear to capture the idea of top-$k$ accuracy, we observe that the analogy fails, as top-$k$ robustness is *not a relaxation* of standard local robustness. For example, if a model is top-1 *accurate*, it is also top-2 accurate; however, if a model is top-1 *robust*, it may not be top-2 robust. Figure 1a provides an example illustrating this point.

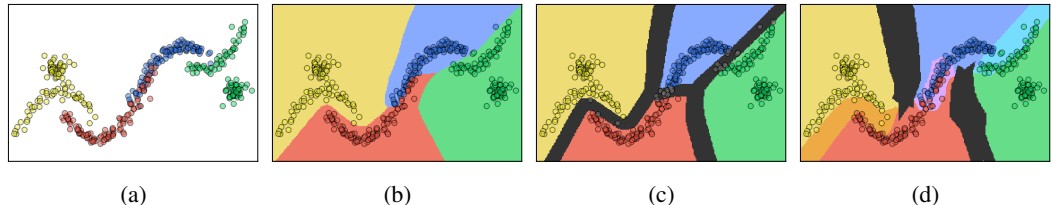

Figure 2: **(a)** An example 2D synthetic dataset containing four classes. **(b)** Decision boundary of a standard model trained on the synthetic dataset. **(c)** Decision boundary of a GloRo Net [22] trained to be certifiably robust on the synthetic dataset. **(d)** Decision boundary of an RT2 GloRo Net (see Section 5) trained to be RT2 robust on the synthetic dataset. We observe that the RT2 GloRo Net can label the points as accurately as the standard model, while the GloRo Net must reject some point on the manifold. The RT2 GloRo Net reports a relaxed robustness guarantee (indicated by orange, purple, and cyan) in the regions where the classes overlap. E.g., in the orange region, the RT2 GloRo Net guarantees that no adversary can change the label to blue or green with a small-norm perturbation.

We thus turn our attention to a modification of Definition 2 that *does* relax standard local robustness. Definition 3 provides a notion of what we call *relaxed* top-$K$ robustness, or RTK robustness, which is properly analogous to the concept of top-$k$ accuracy. Essentially, a point is considered RTK robust if it is top-$k$ robust for some $k$ in $\{1,...,K\}$.

**Definition 3 (Relaxed Top-K Robustness)** *A model, $F$, is relaxed-top-K (RTK) $\epsilon$-locally-robust at point, $x$, w.r.t. norm, $||\cdot||$, if*

$$\forall x' \ . \ ||x - x'|| \leq \epsilon \implies \exists k \leq K : F^k(x) = F^k(x')$$

From the definition, it is clear that RTK robustness is a relaxation of standard local robustness: first, RT1 robustness is equivalent to top-1 robustness, which is equivalent to local robustness; second, RT1 robustness implies RTK robustness for $K > 1$.

We can think of RTK robustness as allowing the model to output a set of labels (with size at most $K$) such that the output set remains invariant under bounded-norm perturbations. In some cases there may be multiple such sets, e.g., if the model is both top-1 robust and top-2 robust on the point (see Figure 1b for an example). These can be thought of as the sets of classes that are "safe" to predict on a given point; if the only such set is empty then no classes are safe, and the model abstains from predicting. This gives rise to a spatially-arranged hierarchy of classes that is conveyed through the robustness guarantee: each region on the decision surface corresponds to some set (or sets) of classes that can be robustly predicted. Figure 2c gives an example of how such a decision surface might look on a synthetic 2D dataset, via a coloring that indicates the *smallest* safe set that can be predicted.

## 4 Affinity Robustness

In Section 3 we demonstrate how to relax local robustness in order to capture a robustness notion analogous to the concept of top-$k$ accuracy. We observed that training for RTK robustness induces a decision surface with a spatially-arranged hierarchy of classes conveyed through the robustness guarantee. While we observed that this hierarchy arises naturally, in some cases we may wish to guide the hierarchy that forms. E.g., we may have *a priori* knowledge of the class hierarchy that we would like to impart to our model, or we may have a limited set of class groupings that are acceptable.

In this section, we demonstrate how the class hierarchy induced on the decision surface of a model can be guided using pre-specified *affinity sets*, which define the sets of classes that can be grouped together without intervening rejected space on the decision surface. More formally, let $\mathcal{S}$ be a collection of affinity sets, where each affinity set $S \in \mathcal{S}$ is simply a set of class labels. We will typically assume that for each class, $j$, $\exists S \in \mathcal{S}$ such that $j \in S$; that is, each class is included in at least one affinity set.

For a given collection of affinity sets, we can define *affinity robustness* (Definition 4), which stipulates that points, $x$, may be top-$k$ robust for any $k$, so long as $F^k(x)$ is contained in some affinity set. That is, classes may be "grouped" together in a robust prediction set with other classes that share an affinity set.

**Definition 4 (Affinity Robustness)** *A model, $F$, is affinity-$\epsilon$-locally-robust on a point, $x$, w.r.t. a collection of affinity sets, $\mathcal{S}$, and norm $||\cdot||$, if*

$$\forall x' \; . \; ||x - x'|| \leq \epsilon \implies \exists k \in \mathbb{N}, S \in \mathcal{S} : F^k(x) = F^k(x') \; \wedge \; F^k(x) \subseteq S$$

**Examples of Affinity Sets.** Several common datasets offer natural instantiations of affinity sets. For example, CIFAR-100 contains 100 classes that are grouped into 20 super-classes containing 5 related classes each. Additionally, Imagenet classes are derived from a tree structure from which many natural collections of affinity sets could be derived. Finally, affinity sets can be used to capture mistakes that may arise even without adversarial manipulation, e.g., by grouping classes that are visually similar; or to group classes that may naturally co-occur in the same instance, e.g., highways and pastures in EuroSAT [11] satellite images (see Section 6.2).

## 5 Efficiently-Certifiable Construction

In this section we describe how to produce networks that incorporate certifiable RTK or affinity robustness into their training objectives. In particular we follow a similar approach to Leino et al. [22], instrumenting the output of a neural network to return an added class, $\bot$, in cases where the desired property cannot be certified. Perhaps surprisingly, our construction demonstrates that our proposed robustness properties can be certified efficiently, *with little overhead compared to a forward pass of the network*.

### 5.1 Background: Globally Robust Neural Networks

Recently, Leino et al. [22] proposed *Globally Robust Neural Networks* (GloRo Nets) for training certifiably robust deep networks, an approach that has been found to be the current state-of-the-art for deterministic $\ell_2$ robustness certification. A GloRo Net encodes robustness certification into its architecture such that all points are either rejected (the network predicts an added class, $\bot$), or certifiably $\epsilon$-locally-robust. GloRo Nets make use of the underlying network's global Lipschitz constant. Specifically, let $f$ be a neural network, let $j$ be the class predicted by $f$ on point, $x$, and let $K_{ji}$ (for $i \neq j$) be the Lipschitz constant of $f_j - f_i$. A GloRo Net adds an extra logit value, $f_\bot(x) = \max_{i \neq j}\{f_i(x) + \epsilon K_{ji}\}$. If the margin between the highest logit output, $f_j(x)$, and the second-highest logit output, $f_i(x)$, is smaller than $\epsilon K_{ji}$, then $f_\bot(x)$ will be the maximal logit score in the GloRo Net, thus the point will be rejected; otherwise the GloRo Net will not predict $\bot$ and the point can be certified as $\epsilon$-locally-robust.

### 5.2 RTK and Affinity GloRo Nets

We propose two variations of GloRo Nets, *RTK GloRo Nets* and *Affinity GloRo Nets*, which naturally satisfy RTK and affinity robustness respectively on all non-rejected points. We first demonstrate how to construct RTK GloRo Nets by instrumenting a model, $f$, such that the instrumented model returns $\bot$ unless $f$ can be certified as RTK $\epsilon$-locally-robust. The construction for Affinity GloRo Nets is similar; it is omitted here, but the details are included in Appendix B in the supplementary material.

As defined in Section 3, let $F$ be the predictions made by $f$, i.e., $F(x) = \text{argmax}_i\{f_i(x)\}$, and let $F^k(x)$ be the set of the top $k$ predictions made by $f$ on $x$; and as above, let $K_{ji}$ be the Lipschitz constant of $f_j - f_i$.

For $k \leq K$ and $j \in F^k(x)$, let $m_j^k(x) = f_j(x) - \max_{i \notin F^k(x)}\{f_i(x) + \epsilon K_{ji}\}$. Intuitively, $m_j^k(x)$ is the margin by which class $j$, which is in the top $k$ classes, exceeds every class not in the top $k$ classes, after accounting for the maximum change in logit values within a radius of $\epsilon$ determined by the Lipschitz constant. We observe that if $m_j^k(x) > 0$ then the logit for class $j$, $f_j(x)$, cannot be surpassed within the $\epsilon$-ball around $x$ by an output not in the top $k$ outputs, $f_i(x)$ for $i \notin F^k(x)$.

Next, let $m^k(x) = \min_{j \in F^k(x)}\{m_j^k(x)\}$. This represents the minimum margin by which *any* class in the top $k$ classes exceeds every class not in the top $k$ classes; thus if $m^k(x) > 0$, then $F$ is top-$k$ robust at $x$.

Finally, let $m(x) = \max_{k \leq K}\{m^k(x)\}$. We observe that if $m(x) > 0$, then the model is RTK robust at $x$. We would thus like to predict $\bot$ only when $m(x) < 0$. To accomplish this we create an instrumented model, $g$, as given by Equation 1. This instrumented model naturally satisfies RTK robustness, as stated by Theorem 1.

$$g_i(x) = f_i(x); \; g_\bot(x) = \max_i\{f_i(x) - m(x)\} \tag{1}$$

**Theorem 1** *Let $g$ be an RTK GloRo Net as defined by Equation 1. Then, if the maximal output of $g(x)$ is not $\perp$, then $F$ is RTK $\epsilon$-locally-robust at $x$.*

The proof of Theorem 1, as well as an analogous theorem stating correctness of our Affinity GloRo Net implementation, is given in Appendix C in the supplementary material.

**Implementation.** In order to implement RTK or Affinity GloRo Nets, we must compute the Lipschitz constant of the instrumented network, $f$. Furthermore, this computation must be differentiable in order to incorporate certification into the learning routine. We approximate an upper bound of the global Lipschitz constant of $f$ by taking a layer-wise product of the spectral norms of each layer's kernel matrix, using the power method. While this is efficient, it may also provide a loose bound leading the GloRo Net to reject more points than necessary. Nonetheless, we find this to be an effective method of certification, as the bound computation is incorporated into the training procedure, allowing the network to learn parameters for which the bound is reasonably tight [22].

For reproducibility and full transparency of our implementation, we have made our code publicly available on GitHub at `https://github.com/klasleino/gloro`.

### 5.3 Other Certification Techniques

While many certification methods have been proposed and studied in the past, we use the GloRo Net approach to certification for two key reasons. First, GloRo Nets have been shown to be among the top state-of-the-art methods for deterministic $\ell_2$ certification, matching or outperforming the VRA of all other deterministic methods recently surveyed by Leino et al. [22]. Second, GloRo Nets provide an elegant way for incorporating our robustness objectives into a *certifiable training procedure*, as certified training can be reduced to simply performing overapproximate certification.

However, leveraging other types of techniques for certifying our proposed robustness properties remains an interesting possibility for future work to explore. Appendix D in the supplementary material provides a discussion of how other certification techniques may be adapted to incorporate RTK and affinity robustness.

## 6 Evaluation

In this section, we motivate our proposed robustness relaxations via an empirical demonstration and argue that our proposed relaxations are likely to be relevant for extending certified defenses to complex prediction tasks with many classes, where standard robustness may be difficult or unrealistic to achieve. To this end, we first find that applying these relaxed robustness variants to suitable domains leads to certifiable models with lower rejection rates and higher certified accuracy (Section 6.1). We then explore the intriguing properties of a model's prediction space that arise when the model is trained with the objective of RTK or affinity robustness on appropriate domains. In particular, we examine the predictions of RTK and Affinity GloRo Nets trained on EuroSAT [11] (Section 6.2) and CIFAR-100 (Section 6.3), and find that (1) the classes that are "grouped" by the model typically follow a logical structure, even without supervision, and (2), affinity robustness can be used to capture a small set of specific, challenging class distinctions that account for a relatively large fraction of the points satisfying RTK robustness but not standard robustness.

In addition, we provide further motivation for Affinity GloRo Nets by showing how they can be leveraged to efficiently certify a previously-studied safety property for ACAS Xu [14], a collision avoidance system for unmanned aircraft that has been a primary motivation in many prior works that study certification of neural network safety properties [10, 15, 16, 23]. Specifically, we show that Affinity GloRo Nets can certify *targeted safe regions* [10] in a single forward pass of the network, while previous techniques based on formal methods require hours to certify this property, even on smaller networks [10]. These experiments are presented in Appendix E in the supplementary material.

### 6.1 Improving Certified Performance through Relaxation

We begin our evaluation by measuring the extent to which certification performance can be improved when the objective is relaxed. We focus particularly on the $\ell_2$ norm, and *deterministic* certification and guarantees, as opposed to the types of guarantees obtained via Randomized Smoothing [3]. To

| dataset | guarantee | $\epsilon$ | VRA / RTK VRA / affinity VRA | rejection rate | clean accuracy* |
|---------|-----------|------------|------------------------------|----------------|-----------------|
| EuroSAT | standard (local robustness) | 0.141 | $0.749 \pm 0.003$ | $0.204 \pm 0.002$ | $0.862 \pm 0.003$ |
| EuroSAT | RT3 | 0.141 | $0.908 \pm 0.002$ | $0.073 \pm 0.002$ | $0.987 \pm 0.002$ |
| EuroSAT | highway+river affinity | 0.141 | $0.798 \pm 0.002$ | $0.170 \pm 0.002$ | $0.917 \pm 0.003$ |
| EuroSAT | highway+river+agriculture affty. | 0.141 | $0.819 \pm 0.003$ | $0.151 \pm 0.003$ | $0.930 \pm 0.003$ |
| CIFAR-100 | standard | 0.141 | $0.281 \pm 0.002$ | $0.640 \pm 0.003$ | $0.473 \pm 0.002$ |
| CIFAR-100 | RT5 | 0.141 | $0.360 \pm 0.002$ | $0.562 \pm 0.002$ | $0.706 \pm 0.003$ |
| CIFAR-100 | superclass affinity | 0.141 | $0.323 \pm 0.002$ | $0.599 \pm 0.002$ | $0.520 \pm 0.002$ |
| Tiny-Imagenet | standard | 0.141 | $0.210^{[22]}$ | $0.639^{[22]}$ | $0.346^{[22]}$ |
| Tiny-Imagenet | RT5 | 0.141 | $0.277 \pm 0.002$ | $0.447 \pm 0.006$ | $0.537 \pm 0.002$ |
| ACAS Xu (App. E) | targeted affinity | 0.010 | $0.749 \pm 0.001$ | $0.195 \pm 0.001$ | $0.858 \pm 0.002$ |

Table 1: Certification results under various notions of robustness. Note that VRA and clean accuracy numbers are given for the VRA/accuracy metric corresponding to the robustness guarantee the respective model was trained for, thus their meaning is slightly slightly different for each guarantee (see Section 6.1:**Metrics**). Results are taken as the average over 10 runs; standard deviations are denoted by $\pm$.

this end, we compare against GloRo Nets [22], which have achieved state-of-the art performance for deterministic certification on several common benchmark datasets.

**Datasets.** Our evaluation focuses on datasets for which our relaxed robustness variants are appropriate. Namely, we select datasets with large numbers of fine-grain classes, or classes with a large degree of feature-overlap: EuroSAT [11], CIFAR-100 [17], and Tiny-Imagenet [19]. The most widely-studied datasets for benchmarking deterministic robustness certification are CIFAR-10 and MNIST, which do not fit these desiderata; however, Tiny-Imagenet has also been used for evaluating deterministic certification [21, 22], making our results directly comparable to the previously-published state-of-the-art.

**Models.** We trained three types of models in our evaluation: GloRo Nets (as a point of comparison to standard robustness certification), RTK GloRo Nets, and Affinity GloRo Nets. More details on the affinity sets chosen for the Affinity GloRo Nets are given in Section 6.2 (for EuroSAT) and Section 6.3 (for CIFAR-100). The details of the architecture, training procedure, and hyperparameters are provided in Appendix F in the supplementary material.

**Metrics.** We measure the performance of RTK models using a metric we call *RTK VRA*, which is the natural analogous metric to top-$k$ accuracy. For a given point, $x$, that is certifiably RTK robust, let $k^*$ be the the maximum $k \leq K$ such that the model is top-$k$ robust at $x$ (recall that an RTK robust point can be top-$k$ robust for more than one $k$—$k^*$ corresponds to the *loosest* such guarantee). We define the RTK VRA of model, $F$, as the fraction of labeled points, $(x,y)$, such that (1) the model is RTK robust at $x$, and (2) $y \in F^{k^*}$. In other words, the correct label must be in a *certifiably-robust set of top-$k$ predictions* for some $k \leq K$. Similarly, we define *affinity VRA*, with respect to a collection of affinity sets, $\mathcal{S}$, as the fraction of points for which the correct label is in a certifiably-robust set of top-$k$ predictions that is contained in some affinity set in $\mathcal{S}$.

We also provide clean accuracy metrics for each model, corresponding to the guarantee the model is trained for; e.g., top-$k$ accuracy for RTK GloRo Nets. For Affinity GloRo Nets, we use what we call *affinity accuracy*, which counts a prediction as correct if all labels scored above the ground truth share a single affinity set with the ground truth.

**Performance.** Table 1 shows the performance of GloRo Nets compared to RTK GloRo Nets and affinity GloRo Nets with respect to the appropriate VRA metric, as well as the rejection rate of each model, i.e., the fraction of points that cannot be certified. RTK GloRo Nets use the most relaxed objective, and accordingly, we see that they consistently outperform the standard GloRo Net, improving VRA performance by 6-16 percentage points. Additionally, RTK GloRo Nets reduce the rejection rate significantly, rejecting as few as half the number of points rejected by the standard GloRo Nets. We also observe that affinity GloRo Nets consistently improve performance compared to standard GloRo Nets. In particular, highway+river+agriculture affinity (see Section 6.2) and superclass affinity (see Section 6.3) increase VRA performance by 8 points and 4 points respectively, despite the fact that these affinity guarantees are significantly more restrictive than the RTK guarantees.

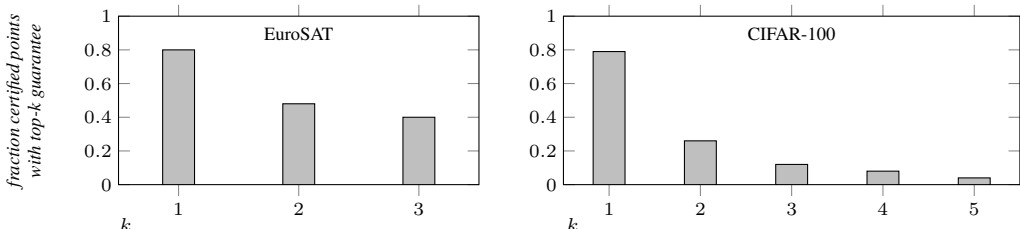

Figure 3: Robust prediction set sizes on EuroSAT (left) and CIFAR-100 (right). The y-axis measures the fraction of certified test instances that were certifiable as top-$k$ robust, for each $k$ on the x-axis. Note that the fractions do not sum to 1 because some points can receive multiple guarantees.

**Relaxed Guarantees & Learning Objectives.** As demonstrated by the results in Table 1, RTK and affinity robustness significantly improve certifiability and VRA performance. Clearly, this improvement is in part due to the relaxed certification and evaluation criteria entailed by RTK and affinity robustness. However, there is also evidence that the relaxed learning objective itself may better aid in learning robust boundaries in general. For example, on CIFAR-100, $5\%$ of points rejected by the GloRo Net are certified by the RT5 GloRo Net with a *top-1 guarantee*, suggesting that the RT5 objective better facilitated obtaining a strong robustness guarantee on these points.

Similarly, we find that the objective of affinity robustness, while technically stronger than RTK robustness, guides the model towards greater certifiability than would be explained by the fraction of instances for which the RTK GloRo Net naturally satisfies the affinity set groupings, highlighting the significance of incorporating affinity robustness into the training objective. For example, on CIFAR-100, $22\%$ of points certified by the RT5 GloRo Net receive a top-$k$ guarantee that does not correspond to a superclass. This would correspond to a $16\%$ higher rejection rate under superclass affinity robustness; meanwhile the rejection rate of the superclass Affinity GloRo Net is in fact only $6\%$ higher than that of the RT5 GloRo Net.

**Top-k Prediction Set Sizes.** While appropriate relaxations are useful in many contexts, it is generally preferable to be able to obtain a *stricter* guarantee—i.e., a *smaller* robust output set. Figure 3 shows the robust prediction set sizes obtained on certified (i.e., non-rejected) test points for both EuroSAT and CIFAR-100. Note that points may receive a top-$k$ guarantee for multiple values of $k$, meaning that a point may belong to robust sets of more than one size. For example, $56\%$ of certified points on EuroSAT and $24\%$ on CIFAR-100 received a guarantee for multiple values of $k$. We see that the majority of points get a tight guarantee, with about $80\%$ of certified points receiving a top-1 guarantee, and looser guarantees becoming progressively less common.

## 6.2 Relaxed Robustness Guarantees on EuroSAT

EuroSAT is a relatively recent dataset based on land-use classification of Sentinel-2 satellite images [11]. Although EuroSAT contains only ten classes, we argue that it is nonetheless a suitable application for the types of relaxed robustness presented in this paper, primarily because of the lack of mutual exclusivity among its classes. Specifically, the classification task proposed for EuroSAT contains the following classes to describe a $64 \times 64$ image patch: (1) annual crop, (2) forest, (3) herbaceous vegetation, (4) highway, (5) industrial buildings, (6) pasture, (7) permanent crop, (8) residential buildings (9) river, and (10) sea/lake. Each instance has exactly one label, however, in practice the labels are not necessarily non-overlapping. For example, highway images may depict a road going through agricultural land, crossing a river, or near buildings. It may be reasonable, then, for a classifier to produce high logit values for two classes simultaneously, making local robustness potentially difficult to achieve.

Indeed, we find this to be the case; namely, many EuroSAT instances labeled "highway" cannot be certified for standard robustness, while a large fraction of these rejected inputs *can* be certified for RT3 robustness. Specifically, we found that a standard GloRo Net trained on EuroSAT rejected $39\%$ of highway instances from the test set (above average for instances overall). Meanwhile, an RT3 GloRo Net was able to certify $72\%$ of these same instances. Upon examination, we found that many of the guarantees associated with the points rejected by the GloRo Net but certified by the RT3 GloRo Net appeal nicely to intuition. Figure 4a depicts a few of these intuitive examples; we see that on image

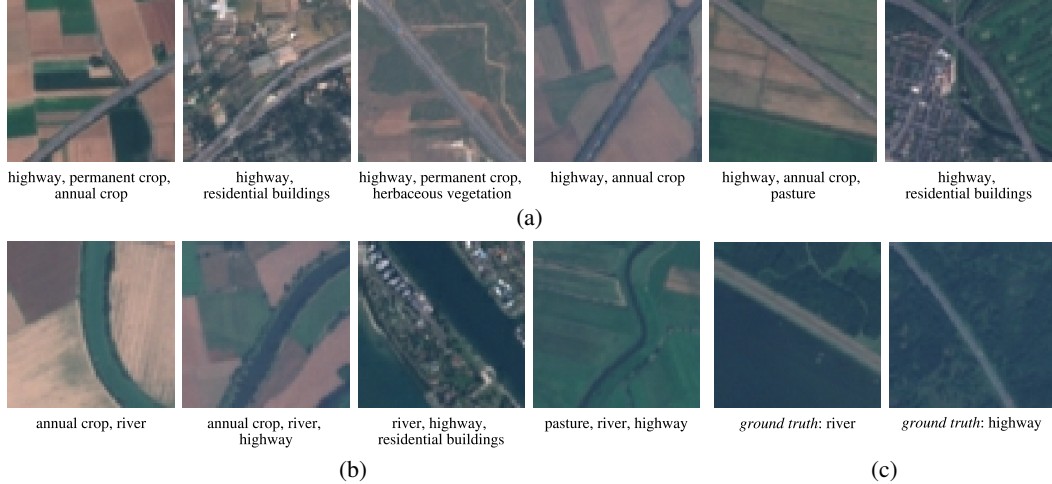

highway, permanent crop, annual crop     highway, residential buildings     highway, permanent crop, herbaceous vegetation     highway, annual crop     highway, annual crop, pasture     highway, residential buildings

(a)

annual crop, river     annual crop, river, highway     river, highway, residential buildings     pasture, river, highway     *ground truth*: river     *ground truth*: highway

(b)                            (c)

Figure 4: Samples of EuroSAT instances labeled "highway" **(a)** or "river" **(b)** that are rejected (i.e., cannot be certified) by a standard GloRo Net, but certified by an RT3 GloRo Net. The classes included in the RT3 robustness guarantee are given beneath each image. **(c)** Two visually similar instances with ground truth label "river" (left) and "highway" (right).

patches with a highway in a field, the RT3 GloRo Net gives top-2 or top-3 guarantees with highway alongside classes such as "annual crop," and on image patches with a highway near a neighborhood, it gives top-2 guarantees with highway grouped with "residential buildings."

We see a similar trend for the "river" class, shown in Figure 4b. Additionally, we find that many instances of rivers receive an RTK guarantee including "highway" as one of the classes belonging to the corresponding robust prediction set. As illustrated by Figure 4c, this may not be unreasonable—Figure 4c shows two visually similar EuroSAT instances with ground truth label "river" and "highway" respectively, demonstrating that the $64 \times 64$ patches may not always provide enough detail and context to easily distinguish these two classes.

The above intuition suggests that the difficulty in learning a certifiably-robust model on EuroSAT may be largely due to frequent cases where specific sets of classes may not be sufficiently separable. That an adversary might have the ability to control which of a set of plausible labels is chosen may be considered inconsequential, provided the adversary cannot cause *arbitrary* mistakes. This observation motivates the use of affinity robustness on EuroSAT. That is, we may wish to further restrict the sets of classes that may forgo $\epsilon$-separation (those that correspond to "arbitrary" mistakes), while at the same time admitting the model to group a specified set of classes, between which adversarial examples could be considered benign.

To this end, we suggest two plausible affinity sets for EuroSAT. The first, which we refer to as *highway+river affinity*, captures the challenges faced by standard GloRo Nets that are illustrated by Figure 4; namely, $\mathcal{S}$ consists of one affinity set, $S_c$ per class, $c$, consisting of $c$, the class "highway," and the class "river." The second, which we refer to as *highway+river+agriculture affinity* additionally allows the classes "permanent crop" and "annual crop" to be grouped, as these classes are often visually similar. We find that these two affinity sets allow us to improve the VRA on EuroSAT compared to a standard GloRo Net by 5 and 7 percentage points, respectively (see Table 1). Moreover, the performance of the highway+river+agriculture Affinity GloRo Net closes half of the VRA gap between the standard GloRo Net and the more-relaxed RTK GloRo Net, suggesting that roughly half of the performance benefits obtained under RTK robustness can be recovered by accounting for a few simple types of reasonable mistakes.

## 6.3 Relaxed Robustness Guarantees on CIFAR-100

CIFAR-100 is another natural application for which our relaxed robustness variants are suitable for several reasons, including that (1) it contains a large number of classes, (2) many of the classes are fine-grain, especially considering the small size of the input images, and (3) its 100 classes are further organized into 20 *superclasses* that each encompass 5 classes.

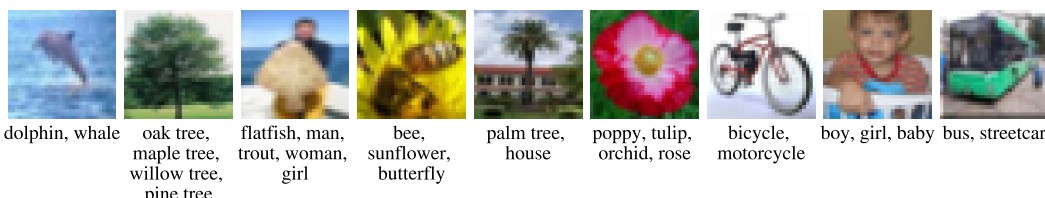

| dolphin, whale | oak tree, maple tree, willow tree, pine tree | flatfish, man, trout, woman, girl | bee, sunflower, butterfly | palm tree, house | poppy, tulip, orchid, rose | bicycle, motorcycle | boy, girl, baby | bus, streetcar |

Figure 5: Samples of CIFAR-100 instances that are both correctly classified and certified as top-$k$ robust for $k > 1$. The classes included in the RT5 robustness guarantee are given beneath each image.

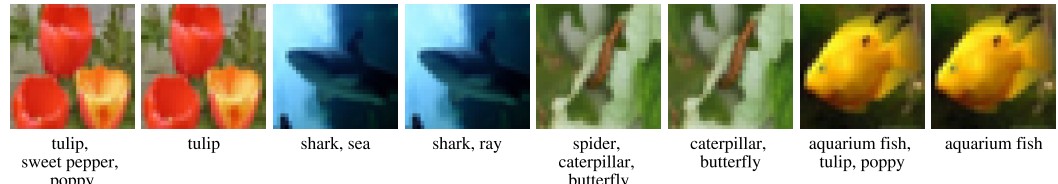

| tulip, sweet pepper, poppy | tulip | shark, sea | shark, ray | spider, caterpillar, butterfly | caterpillar, butterfly | aquarium fish, tulip, poppy | aquarium fish |

Figure 6: Comparison of robust prediction sets produced by an RT5 GloRo Net (left) and a superclass Affinity GloRo Net (right). Samples are taken from points on which the RT5 GloRo Net did not match a single superclass, while the Affinity GloRo Net was able to successfully certify the point.

We find that when training for RT5 robustness using an RT5 GloRo Net, $78\%$ of certifiable points on the RT5 GloRo Net have corresponding robust prediction sets that are contained by some superclass set. That is, even without supervision, the robust prediction sets of the RTK GloRo net typically respect the superclass hierarchy. Moreover, even on instances for which the robust prediction set does not match a superclass set, the robust prediction set is often nonetheless "reasonable," in that it is often clear upon inspection why the model may have chosen the particular set of predictions. Figure 5 provides samples of correctly-classified, RT5-certifiable points with their corresponding robust prediction sets, illustrating this point. More such examples can be found in Appendix G in the supplementary material.

However, supposing we would want to strictly enforce the model's robust prediction sets to be contained entirely in one superclass, Affinity GloRo Nets provide a means of doing this. We find that $34\%$ of instances on which the RT5 GloRo Net fails to match a superclass can be certified by the superclass Affinity GloRo Net. Figure 6 provides examples of such instances, showing that the additional supervision of Affinity GloRo Nets helps better ensure the robust prediction sets respect superclasses.

## 7  Conclusion

In this work, we introduce two novel safety properties for classifiers that relax local robustness in order to provide a more practical objective for certifiable defenses in complex prediction tasks where standard robustness may be difficult or unrealistic to achieve. The first property, RTK robustness, constitutes the first robustness notion extending to top-$k$ prediction tasks that can be certified without knowledge of the ground-truth label. The second, affinity robustness, is a novel robustness notion tailored to certifiable defenses against targeted adversarial examples. We show how to construct models that can be efficiently certified against each relaxed robustness property, and demonstrate that these properties are well-suited to several significant classification problems, leading to lower rejection rates and higher certified accuracies than can be obtained when certifying "standard" local robustness. We suggest that this work will be useful in striving towards performance parity between certifiable and non-certifiable classification; and more generally, that broader, domain-appropriate safety guarantees should be considered for certifying model safety. Finally, we note that although robustness certification is typically helpful in making machine learning safe for high-stakes contexts, techniques drawn from evasion attacks may be used to protect privacy [26], meaning robust models may thwart such avenues for anonymity.

## Acknowledgments and Disclosure of Funding

The work described in this paper has been supported by the Software Engineering Institute under its FFRDC Contract No. FA8702-15-D-0002 with the U.S. Department of Defense, and by the National Science Foundation under Grant No. CNS-1943016.

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
