# Supplementary Material: Relaxing Local Robustness

**Klas Leino**
Carnegie Mellon University
kleino@cs.cmu.edu

**Matt Fredrikson**
Carnegie Mellon University
mfredrik@cs.cmu.edu

## A   Discussion of the Method Proposed by Jia et al.

In recent work, Jia et al. [4] also proposed an approach that aims to capture certified robustness for top-$k$ predictions. In this section, we provide details on differences between this work and ours; particularly the shortcomings of this approach that our work addresses.

Recall our notation from Section 3, where, for neural network, $f$, we define $F^k(x)$ as the set of classes corresponding to the top $k$ logit values in $f(x)$. The approach of Jia et al. provides a *probabilistic* bound on the radius, $r_j^k(x)$, under which a given class $j \in F^k(x)$ will remain in the top-$k$ predictions. That is, Jia et al. provide a probabilistic guarantee that Equation A1 holds.

$$\forall x' \ . \ ||x - x'|| \leq r_j^k(x) \implies j \in F^k(x') \tag{A1}$$

In their evaluation, Jia et al. consider a point, $x$, to be *certified* at radius, $\epsilon$, if $r_{y^*}^k \geq \epsilon$, where $y^*$ is the *ground truth* label for $x$. This presents a problem for certifying unseen points as the ground truth cannot be known. We therefore stipulate that certification must be independent of the true label of the point being certified. Moreover, replacing the ground truth with the predicted label is unsatisfactory, because the purpose of generalizing to top-$k$ predictions is to consider cases where *any* of the predictions in $F^k(x)$ may be correct.

While Jia et al. do not address this issue, one straightforward adaptation of their approach is to take the minimum certified radius over all classes in $F^k(x)$. That is, let $r^k(x) = \min_{j \in F^k(x)}\{r_j^k(x)\}$. We see that this leads to a natural guarantee given by Equation A2, which can be certified without knowledge of the ground truth class. In short, Equation A2 holds because by taking the minimum $r_j^k$, we are guaranteed that *all* of the top $k$ classes will remain in the top $k$ classes under perturbations bounded by $r^k(x)$.

$$\forall x' \ . \ ||x - x'|| \leq r^k(x) \implies F^k(x) = F^k(x') \tag{A2}$$

We note however, that when certification is determined by comparing $r^k(x)$ to a fixed radius, Equation A2 is equivalent to our proposed definition for *top-$k$ robustness* (Definition 2). As discussed in Section 3, this is also *not* a satisfactory robustness analogue to top-$k$ accuracy as it does not relax local robustness. We therefore argue that *RTK robustness* (Definition 3) should be used to certify the robustness of top-$k$ predictions.

## B   Construction of Affinity GloRo Nets

In this section we describe how to produce networks that incorporate certifiable affinity robustness into their training objectives. As with the construction for RTK GloRo Nets in Section 5, we follow a similar approach to Leino et al. [9], instrumenting the output of a neural network to return an added class, $\perp$, in cases where affinity robustness cannot be certified. To this end, we propose *Affinity GloRo Nets*, which naturally satisfy affinity robustness on all non-rejected points.

As defined in Section 3, let $F$ be the predictions made by $f$, i.e., $F(x) = \text{argmax}_i\{f_i(x)\}$, and let $F^k(x)$ be the set of the top $k$ predictions made by $f$ on $x$; and as above, let $\mathcal{K}_{ji}$ be the Lipschitz constant of $f_j - f_i$. Furthermore, let $\mathcal{S}$ be a collection of affinity sets, and let $K = \max_{S \in \mathcal{S}}\{|S|\}$; that is, $K$ is the size of the largest affinity set in $\mathcal{S}$.

35th Conference on Neural Information Processing Systems (NeurIPS 2021).

For $k \leq K$ and $j \in F^k(x)$, let $m_j^k(x) = f_j(x) - \max_{i \notin F^k(x)}\{f_i(x) + \epsilon \mathcal{K}_{ji}\}$. Intuitively, $m_j^k(x)$ is the margin by which class $j$, which is in the top $k$ classes, exceeds every class not in the top $k$ classes, after accounting for the maximum change in logit values within a radius of $\epsilon$ determined by the Lipschitz constant. We observe that if $m_j^k(x) > 0$ then the logit for class $j$, $f_j(x)$, cannot be surpassed within the $\epsilon$-ball around $x$ by an output not in the top $k$ outputs, $f_i(x)$ for $i \notin F^k(x)$.

Next, let $m^k(x) = \min_{j \in F^k(x)}\{m_j^k(x)\}$. This represents the minimum margin by which *any* class in the top $k$ classes exceeds every class not in the top $k$ classes; thus if $m^k(x) > 0$, then $F$ is top-$k$ robust at $x$.

Finally, let $m(\mathcal{S}, x)$ be given by Equation B1. Essentially, we restrict our consideration of sets of top-$k$ predictions to those that are constrained to a single affinity set. Among the considered sets, we take the maximum margin by which every class in the set will surpass every class not in the set under bounded perturbations to $x$.

$$m(\mathcal{S}, x) = \max_{k \,:\, \exists S \in \mathcal{S} \,:\, F^k(x) \subseteq S} \left\{ m^k(x) \right\} \tag{B1}$$

We note that in practice, the maximum in Equation B1 can be computed efficiently by representing the sets $F^k(x)$ and $S$ as bit maps and masking out rows of $m^k$ that correspond to values of $k$ for which $F^k(x) \cap S \neq F^k(x)$ for all $S \in \mathcal{S}$.

We observe that if $m(\mathcal{S}, x) > 0$, then the model is affinity robust at $x$. We would thus like to predict $\perp$ only when $m(\mathcal{S}, x) < 0$. To accomplish this we create an instrumented model, $g$, as given by Equation B2.

$$g_i(x) = f_i(x); \quad g_\perp(x) = \max_i\{f_i(x) - m(\mathcal{S}, x)\} \tag{B2}$$

## C  Proofs

### C.1  Correctness of RTK GloRo Nets

**Theorem 1.** *Let $g$ be an RTK GloRo Net as defined by Equation 1. Then, if the maximal output of $g(x)$ is not $\perp$, then $F$ is RTK $\epsilon$-locally-robust at $x$.*

*Proof.* Let $y = F(x) = \operatorname{argmax}_i\{f_i(x)\}$. Assume that the maximal output of $g(x)$ is not $\perp$, i.e., $\exists i$ such that $g_\perp(x) < g_i(x) \leq g_y(x) = f_y(x)$. By the definition of $g_\perp$ in Equation 1, we obtain (C1). By the definition of $y$, we obtain (C2).

$$f_y(x) > \max_i\{f_i(x) - m(x)\} \tag{C1}$$

$$= f_y(x) - m(x) \tag{C2}$$

Thus, we have that $m(x)$ is positive. As $m(x)$ is defined as $\max_{k \leq K}\{m^k(x)\}$, this means that there exists some $k^* \leq K$ such that $m^{k^*}(x) > 0$ (C3).

We recall from the definition of RTK robustness, we must show that there exists some $k \leq K$ such that for all $x'$ at distance no greater than $\epsilon$ from $x$, $F^k(x) = F^k(x')$. We proceed to show that $k^*$ is such a $k$; i.e., $||x - x'|| \leq \epsilon \implies F^{k^*}(x) = F^{k^*}(x')$.

From (C3) we expand the definition of $m^k(x)$ to obtain (C4); and the definition of $m_j^k$ to obtain (C5).

$$0 < m^{k^*}(x) \tag{C3}$$

$$= \min_{j \in F^{k^*}(x)} \left\{ m_j^{k^*}(x) \right\} \tag{C4}$$

$$= \min_{j \in F^{k^*}(x),\, i \notin F^{k^*}(x)} \left\{ f_j(x) - f_i(x) - \epsilon \mathcal{K}_{ji} \right\} \tag{C5}$$

We observe that (C5) implies (C6).

$$\forall j \in F^{k^*}(x),\, i \notin F^{k^*}(x) \; . \; f_i(x) + \epsilon \mathcal{K}_{ji} < f_j(x) \tag{C6}$$

Next, we assume $x'$ satisfies $||x - x'|| \leq \epsilon$. As $\mathcal{K}_{ji}$ is an upper bound on the Lipschitz constant of $f_j - f_i$ we obtain (C7).

$$\frac{|f_j(x) - f_i(x) - (f_j(x') - f_i(x'))|}{||x - x'||} \leq \mathcal{K}_{ji}$$
$$\implies |f_j(x) - f_i(x) - (f_j(x') - f_i(x'))| \leq \mathcal{K}_{ji}\epsilon \qquad \text{(C7)}$$

Thus we argue as follows for all $j \in F^{k^*}(x)$ and $i \notin F^{k^*}(x)$. First, by applying (C7), we obtain (C8). Then, by applying (C6) we obtain (C9).

$$f_i(x) + f_j(x) - f_i(x) - f_j(x') + f_i(x')$$
$$\leq f_i(x) + |f_j(x) - f_i(x) - f_j(x') + f_i(x')|$$
$$\leq f_i(x) + \epsilon\mathcal{K}_{ji} \qquad \text{(C8)}$$
$$< f_j(x) \qquad \text{(C9)}$$

By rearranging the terms in the above inequality, we obtain (C10).

$$\forall j \in F^{k^*}(x), \; i \notin F^{k^*}(x) \; . \; f_i(x') < f_j(x') \qquad \text{(C10)}$$

Finally, we realize that (C10) is equivalent to $F^{k^*}(x) = F^{k^*}(x')$. To see why, consider the following. Let $Z = \{\forall i \; . \; f_i(x')\}$ be the set of all logit values produced by $f$ on $x'$ (assume WLOG each logit value is unique), and let $Z_{k^*} = \{\forall j \in F^{k^*}(x) \; . \; f_j(x')\}$ be the subset of $Z$ containing the logit values of $f(x')$ corresponding to the classes in $F^{k^*}(x)$. By (C10) we have that for all $z \in Z_{k^*}$ and $z' \in Z \setminus Z_{k^*}, z > z'$. Thus, $Z_{k^*}$ contains the top $k^*$ elements of $Z$.

Putting everything together, we conclude that $\exists k \leq K \; : \; ||x - x'|| \leq \epsilon \implies F^k(x) = F^k(x')$. $\quad\square$

### C.2 Correctness of Affinity GloRo Nets

**Theorem 2.** *Let $g$ be an Affinity GloRo Net as defined by Equation B2. Then, if the maximal output of $g(x)$ is not $\perp$, then $F$ is affinity $\epsilon$-locally-robust at $x$.*

*Proof.* The proof follows a similar approach to the proof of Theorem 1. Let $y = F(x) = \text{argmax}_i\{f_i(x)\}$. Assume that the maximal output of $g(x)$ is not $\perp$, i.e., $\exists i$ such that $g_\perp(x) < g_i(x) \leq g_y(x) = f_y(x)$. By the definition of $g_\perp$ in Equation B2, we obtain (C11). By the definition of $y$, we obtain (C12).

$$f_y(x) > \max_i\{f_i(x) - m(\mathcal{S}, x)\} \qquad \text{(C11)}$$
$$= f_y(x) - m(\mathcal{S}, x) \qquad \text{(C12)}$$

Thus, we have that $m(\mathcal{S}, x)$ is positive. From the definition of $m(\mathcal{S}, x)$ in Equation B1, we conclude that there exists some $k^*$ and some $S^* \in \mathcal{S}$, such that $F^{k^*}(x) \subseteq S^*$ and $m^{k^*}(x) > 0$.

We recall from the definition of affinity robustness, we must show that there exists some $k$ and $S$ such that for all $x'$ at distance no greater than $\epsilon$ from $x$, $F^k(x) = F^k(x')$ and $F^k(x) \subseteq S$. We proceed to show that $||x - x'|| \leq \epsilon \implies F^{k^*}(x) = F^{k^*}(x') \; \wedge \; F^{k^*}(x) \subseteq S^*$.

From our observations above, we have that $F^{k^*}(x) \subseteq S^*$, therefore it suffices to simply show that $||x - x'|| \leq \epsilon \implies F^{k^*}(x) = F^{k^*}(x')$, given that $m^{k^*}(x) > 0$. As $m^k(x)$ is defined the same for both Affinity GloRo Nets (Equation B2) and RTK GloRo Nets (Equation 1), the remainder of the proof proceeds exactly as the proof for Theorem 1 in Section C.1. $\quad\square$

## D Discussion of Other Certification Techniques

While many certification methods have been proposed and studied in the past, we use the GloRo Net approach to certification for two key reasons. First, GloRo Nets have been shown to be among the top state-of-the-art methods for deterministic $\ell_2$ certification, matching or outperforming the VRA of all other deterministic methods recently surveyed by Leino et al. [9]. Second, GloRo Nets provide an elegant way for incorporating our robustness objectives into a *certifiable training procedure*, as certified training can be reduced to simply performing overapproximate certification.

However, leveraging other types of techniques for certifying our proposed robustness properties remains an interesting possibility for future work to explore. We now present a discussion of how other certification techniques in the literature may be adapted to incorporate RTK and affinity robustness.

**Randomized Smoothing.** Jia et al. [4] suggest a technique based on randomized smoothing that is in the same spirit as RTK robustness. While we point out several problems with their approach, Appendix A describes how to adapt their technique to get an equivalent property to top-k robustness (Definition 2). However, as we argue in Section 3, RTK robustness (Definition 3) is the correct analogue to top-k accuracy.

In general, RTK robustness can be certified given a method to certify top-k accuracy, by iteratively checking whether top-$k$ robustness holds for any $k$ in $\{1, ..., K\}$. Thus, it should be possible to adapt the method of Jia et al. to correctly certify RTK robustness via randomized smoothing. A naive implementation of this would be very expensive (following the approach outlined in Appendix A), however, as it would require $O(K^2)$ calls to Jia et al.'s method, which is already expensive. On hardware similar to our own, we estimate such an implementation would take about a minute to evaluate and certify each instance[1]. Future work may be able to determine a more efficient way to achieve this.

Finally, we note that although randomized smoothing can, in theory, obtain certificates without any requirements on the original model, in order to get reasonable performance, it is necessary to augment the data with Gaussian noise during training—this allows the model to behave well under the noise that will be introduced at evaluation time. It is therefore possible that for similar reasons, additional training considerations would be important to make use of the relaxed guarantee in smoothed models.

**1-Lipschitz Models.** Recently, a few similar certification approaches to GloRo Nets have been proposed, which use orthonormal projections to ensure the model is 1-Lipschitz [11]. When the model is 1-Lipschitz, certification simply requires a margin of $\epsilon$ between the top logit output and the others. Because these techniques achieve robustness through Lipschitzness, the certification aspect of the method described in Section 5 would also apply to such methods.

Primarily, these approaches differ from GloRo Nets in the way they are *trained*, targeting robustness though a hinge-like loss function that encourages a sufficiently large margin between classes. Because of the nuances of RTK robustness, it is less clear how RTK robustness could be achieved with a simple loss function (e.g., RTK robustness is a naturally disjunctive property, and the network should have a choice in which $k$ to target). However, the GloRo training we propose avoids the challenges of redesigning a proper loss function, and would still work to train models with orthonormalized kernels.

**Convex Relaxation/Bound Propagation.** It is not immediately clear to us how to achieve RTK (or Affinity) robustness with approaches like KW [12] or IBP-based methods [8], but we believe this could be an interesting future direction. It may be possible to change these methods to ensure a margin between the $k^{\text{th}}$ class and the $(k+1)^{\text{th}}$ class. This in turn could be used as a step in certifying RTK robustness as noted in our discussion of Randomized Smoothing. Note, however, that this would likely mean that KW and IBP would require multiple propagations to handle RTK robustness, making them more expensive (while the analysis of the GloRo approach only needs to consider the logits, which only need to be computed once).

An additional important issue, moreover, is incorporating RTK robustness into the learning objective. Because of the disjunctive nature of RTK robustness, it is not obvious how to design the loss function to promote it. GloRo Nets avoid this issue because they essentially make an equivalence between the problem of certification and the robust learning objective.

# E   Relaxed Robustness Guarantees on ACAS Xu

We provide further motivation for Affinity GloRo Nets by showing how they can be leveraged to efficiently certify a previously-studied safety property for ACAS Xu [5], a classification problem

---

[1]this computation would require evaluating the underlying model on 100,000 samples (the number of samples required for adequate smoothing [2]) $K^2$ times

| dataset | guarantee | $\epsilon$ | VRA | rejection rate | clean accuracy |
|---------|-----------|------------|-----|----------------|----------------|
| ACAS Xu | targeted affinity | 0.010 | $0.749 \pm 0.001$ | $0.195 \pm 0.001$ | $0.858 \pm 0.002$ |

Table E1: Certification results under affinity robustness on ACAS Xu. VRA is given as the fraction of points that are correctly classified and affinity robust. Results are taken as the average over 10 runs; standard deviations are denoted by $\pm$.

that has been a primary motivation in many prior works that study certification of neural network safety properties [6, 7, 10, 3]. Specifically, we show that Affinity GloRo Nets can certify *targeted safe regions* [3] in a single forward pass of the network, while previous techniques based on formal methods require hours to certify this property, even on smaller networks [3]. We present these experiments here.

ACAS Xu is an airborne collision avoidance system for unmanned aircraft that has been studied in the context of neural network safety certification. The classification task for ACAS Xu is as follows. Given a few features, e.g., altitude, velocity, etc., the network produces a horizontal maneuver advisory for the aircraft, instructing it on how to turn to avoid a collision. This advisory comes from one of the following options: (1) hard left, (2) left, (3) clear of conflict (i.e., go straight), (4) right, or (5) hard right.

Access to the ACAS Xu dataset is not public. However, many trained networks that have been certified for other safety properties specific to ACAS Xu (and unrelated to robustness) are publicly available; we generated a synthetic dataset derived from the predictions of one such public model provided by Katz et al. [6]. To create this dataset, we generated random inputs clipped to be within the standard range for each input [6] and labeled them using a publicly-available pretrained ACAS Xu network.

We chose our value for $\epsilon$ by estimating the minimum $\ell_2$ distance between any two points with different labels. This value was approximately $0.02$ on our synthetic dataset; thus we used $\epsilon = 0.01$, as $\epsilon$-local-robustness requires a separation of $2\epsilon$ between classes.

Previously, Gopinath et al. [3] proposed certification of *targeted safe regions* on ACAS Xu. An $\ell_p$ ball with radius $\epsilon$, centered at $x$, is considered *targeted safe* if the horizontal maneuver advisory does not change within the ball except to either one degree further left or one degree further right. E.g., an $\ell_p$ ball may contain the directives "left" and "hard left" or "left" and "clear of conflict." We note that this property can be captured by affinity robustness, where the affinity sets are simply all pairs of adjacent directives.

Table E1 presents the results of training and evaluating an Affinity GloRo Net on our synthetic ACAS Xu dataset. In particular, we give the VRA (as the fraction of points that are correctly classified and affinity robust), the rejection rate (as the fraction of points that cannot be certified as affinity robust), and the clean accuracy. Gopinath et al. do not present any directly-comparable metrics to these in their evaluation. Rather than presenting rejection rates for a fixed epsilon, they instead attempt to determine the minimum radius under which the property holds, and they present the average such radius. It is therefore unclear what fraction of points would be accepted under any fixed radius. However, $16\%$ of the points timed out after 12 hours, meaning that *at least* $16\%$ of points were unable to be certified. This is comparable to the $19\%$ rejection rate obtained by our Affinity GloRo Net. Moreover, the points that were successfully certified by Gopinath et al. took, on average, 7.6 hours to certify; by comparison, our approach certifies points in a single forward pass of the network, meaning *an entire batch* can be certified *in a matter of milliseconds*.

# F   Details on Hyperparameters

In this section we provide the details on the architecture, training procedure, hyperparameters, etc., used to train the models in our evaluation.

**Hardware.**   All experiments were run on an NVIDIA TITAN RTX GPU with 24 GB of RAM, and a 4.2GHz Intel Core i7-7700K with 32 GB of RAM.

| dataset | guarantee | epochs | batch size | learning rate schedule | loss function | TRADES schedule |
|---------|-----------|--------|-----------|------------------------|---------------|-----------------|
| EuroSAT | standard | 200 | 256 | $10^{-3} \to 10^{-6}$ after half | TRADES | $0.01 \to 1.2$ log half |
| EuroSAT | RT3 | 200 | 256 | $10^{-3} \to 10^{-6}$ after half | TRADES | $1.0 \to 1.2$ |
| EuroSAT | highway+river affinity | 200 | 256 | $10^{-3} \to 10^{-6}$ after half | cross-entropy | N/A |
| EuroSAT | highway+river+agriculture affty. | 200 | 256 | $10^{-3} \to 10^{-6}$ after half | cross-entropy | N/A |
| CIFAR-100 | standard | 200 | 256 | $10^{-3} \to 10^{-6}$ after half | TRADES | $0.01 \to 1.2$ log half |
| CIFAR-100 | RT5 | 200 | 256 | $10^{-3} \to 10^{-6}$ after half | TRADES | $0.01 \to 1.2$ log half |
| CIFAR-100 | superclass affinity | 200 | 256 | $10^{-3} \to 10^{-6}$ after half | TRADES | $0.01 \to 1.2$ log half |
| Tiny-Imagenet | RT5 | 200 | 256 | $2.5 \cdot 10^{-5} \to 5 \cdot 10^{-6}$ after half | TRADES | $1.0 \to 10.0$ half |
| ACAS Xu | targeted affinity | 100 | 128 | $10^{-3} \to 5 \cdot 10^{-6}$ after half | cross-entropy | N/A |

Table F1: Hyperparameters for each model used in the evaluation in Section 6.

**Data Splits.** On CIFAR-100 and Tiny-Imagenet, we used the standard train-test split. We are unaware of any "standard" train-test split for EuroSAT; thus we used $2/3$ of the data for the training set and $1/3$ of the data for the test set.

**Architectures.** For CIFAR-100 and EuroSAT we used a simple convolutional architecture consisting of two blocks with width 32 and 64 respectively—each containing a convolutional layer with $3 \times 3$ followed by a down-sampling layer—followed by two dense hidden layers with width 256 each. For Tiny-Imagenet we used the same architecture as was used previously by Lee et al. [8] and subsequently Leino et al. [9]. This architecture consists of three convolutional blocks followed by a dense layer of width 256. The first block is composed of two convolutional layers with $3 \times 3$ filters and 64 channels followed by a strided convolution with $4 \times 4$ filters and 64 channels. The first block is the same as the first block, but with a width of 128. The third block has one convolutional layer with $3 \times 3$ filters and 256 channels followed by a strided convolution with $4 \times 4$ filters and 256 channels. For ACAS Xu, we used a dense network consisting of three hidden layers with 1,000 neurons each. In all networks, we used *min-max* activations [1] rather than ReLU activations, as these have been observed to achieve better performance with GloRo Nets [9]. Similarly, down-sampling in the CIFAR-100 and EuroSAT models was achieved via *invertible down-sampling* [1] rather than max-pooling.

**Hyperparameters.** Details on the hyperparameters used to train each model presented in the evaluation in Section 6 are given in Table F1. All models were trained using the ADAM optimizer.

As briefly discussed in Section 5, GloRo Nets (and our variations thereof) make use of the underlying model's Lipschitz constant, which is approximated using the power method (see [9] for more details). Prior to evaluation, the power method is run to convergence, however, during training we run a fixed number of iterations on each batch. For each of the models in our evaluation we used two power iterations on each training batch.

We used a continuous learning rate schedule, where the learning rate was adjusted on each epoch. A description of the learning rate schedule is given in Table F1. E.g., learning rate schedule of "$10^{-3} \to 10^{-6}$ after half" means that the learning rate begins at $10^{-3}$ and halfway through training the learning rate decays exponentially to reach $10^{-6}$ on the final epoch.

For each model, we used one of two loss functions adapted for GloRo Nets as proposed by Leino et al. [9], specified in the "loss function" column of Table F1: *cross-entropy* or *TRADES* (see [9] for more details on these loss functions). The TRADES loss function takes an additional hyperparameter, $\lambda$, which was scheduled during training. Where applicable, the "*TRADES schedule*" column describes how this parameter was scheduled over training. For example, "$0.01 \to 1.2$ log half" means that $\lambda$ was initialized to 0.1 and was increased logarithmically each epoch (i.e., it increases at a decreasing rate) to reach a final value of 1.2 halfway through training; and "$1.0 \to 1.2$" means that $\lambda$ was initialized to 1.0 and was increased linearly to reach 1.2 on the final epoch.

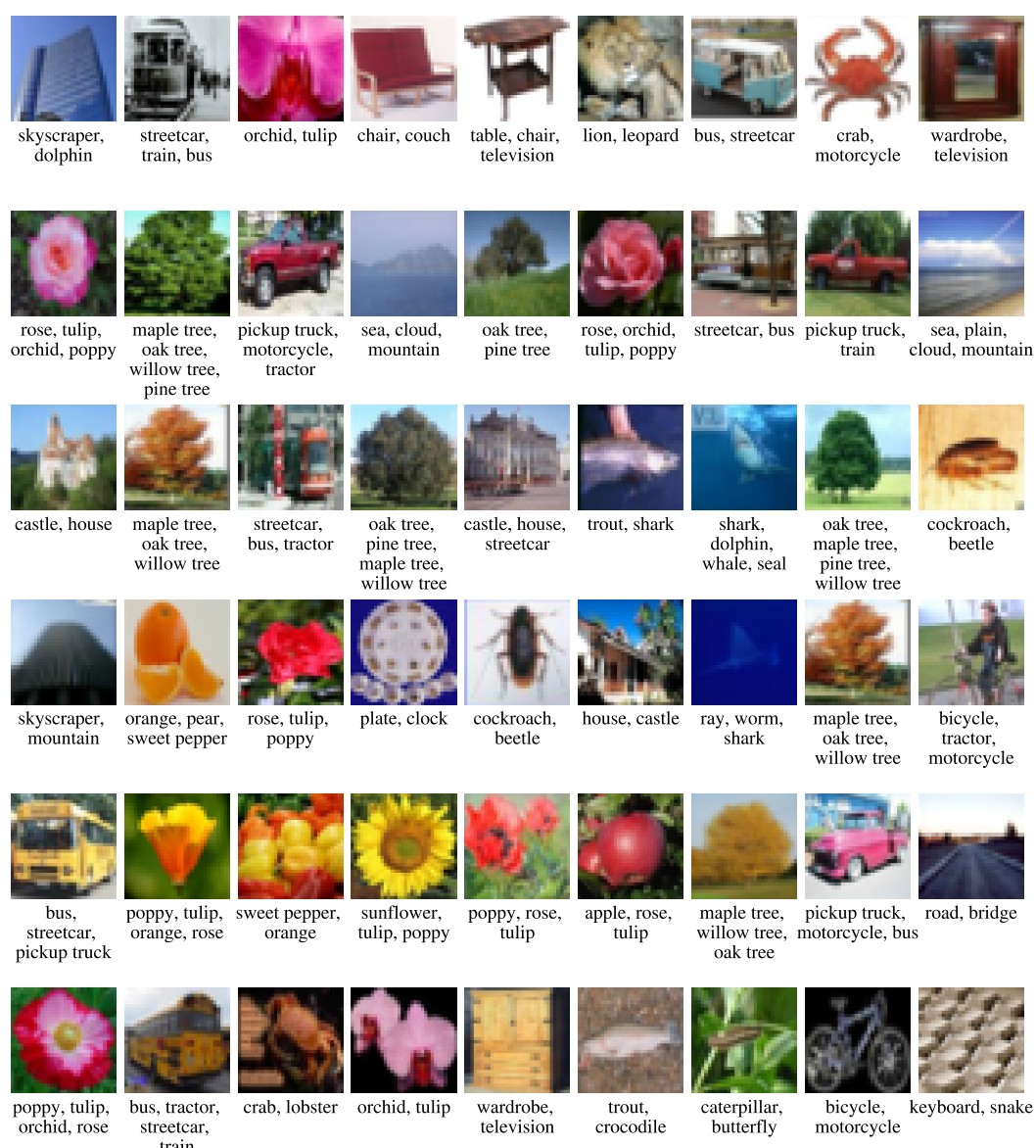

Figure G1: Random samples of CIFAR-100 instances that are both correctly classified and certified as top-$k$ robust for $k > 1$. The classes included in the RT5 robustness guarantee are given beneath each image.

# G   Further CIFAR-100 Results

**Additional Samples of RT5-Certifiable Points.**   In Section 6.3, we provide a few selected samples of correctly-classified, RT5-certifiable points with their corresponding robust prediction sets (Figure 5). Figure G1 provides a larger random sample of such examples, illustrating the types of non-top-1 guarantees that are typically achieved by an RT5 GloRo Net on CIFAR-100.

**Comparing Top-k Relaxations to Reducing the Number of Classes.**   To an extent, relaxations like RTK robustness are somewhat similar to considering a classification problem with fewer unique classes. We note, however, that achieving RTK VRA is still more difficult than achieving the same standard VRA with $1/K$ the number of classes. This is because the loosest top-$k$ guarantee on a given point might be some $k < K$, in which case the correct class for that instance would need to be among the top $k$ classes rather than the top $K$ (see Section 6.1:**Metrics** for details on how RTK and

standard VRA are computed). E.g., some points receive only a top-1 guarantee even under a relaxed robustness variant, meaning that for those points to count towards the RTK VRA, they would need to be classified *exactly correctly*, not simply result in the correct label appearing among the top $K$ outputs.

For the sake of examining this juxtaposition, we trained a network on CIFAR-100 for RT10 robustness in order to compare against state-of-the-art results on CIFAR-10 (which has similar images and $1/\kappa$ the number of classes). Leino et al. [9] find the state-of-the-art standard VRA on CIFAR-10 with radius of $\epsilon = 0.141$ to be approximately 58%. Meanwhile, we were able to achieve approximately 55% RT10 VRA on CIFAR-100 with the same radius. Thus, these findings are essentially in line with the intuition that RTK robustness is similar to, but slightly more challenging than the objective of standard robustness with $1/\kappa$ the number of classes.