# OpenReview forum: "Relaxing Local Robustness"
_NeurIPS.cc/2021/Conference — NeurIPS 2021 Poster_

### Official Review · Reviewer_4qXm · 2021-07-13

**Rating:** 6
**Confidence:** 2

**Summary:**

This paper describes a relaxation of local robustness to top-k robustness and affinity robustness. In addition, the authors connect the relaxation properties with the globally robust neural networks for certification. In the experimental results, the authors found lower rejection rates and higher certified accuracy in a few domains.


**Ethics Review Area:**

["I don’t know"]

**Limitations And Societal Impact:**

It would be nice to explicitly talk about the limitation of the approach. For example, I wonder whether top-k robustness is too restricted (same top-k predictions) and whether other alternatives could be better. I am also concerned with the scalability of the certified models. Currently all the evaluated datasets and models are still quite small. Besides GloRo Nets, could the proposed properties be applied to other types of certified networks or certified training?

**Main Review:**

- Originality:
  * This work introduces new concepts of top-k robustness, relaxed top-k robustness, and affinity robustness, which seems novel for robustness in some domains (e.g. medical prediction).
The presented analysis on certification performance with RTK, affinity is also quite interesting.

- Quality:
  * In a way the main contribution of the work is the introduction of the relaxed robustness metrics, and extension to the GloRo Nets. Given such a new framework, there are less previous work for comparison. I wonder about the tightness and constraints of the proposed certification models, which is not discussed too much in the paper. I also wonder whether the proposed approach can be extended to other types of verifications (such as randomized smoothing, bound propagation).

- Clarity:
  * I found that the clarity of sections 5.1 and 5.2 can be improved. It is a bit dense and harder to read compared to other sections.
  * It is not clear to me why the proposed approaches could improve clean accuracy over the standard guarantee.

- Significance:
  * This work could be an interesting starting point for considering beyond conventional local robustness setup. The authors have shown interesting analysis and improved certificated accuracy / lower rejection rate compared to standard setup.

**Time Spent Reviewing:**

2.25

---

> ### Author Response · Authors · 2021-08-09
> **Thank you for your thoughtful review!**
>
> Thank you for your thoughtful review! We have replied to your specific comments below, and will aim to clarify these points in the final version of the text as well.
>
> > I wonder about the tightness and constraints of the proposed certification models
>
> While potential looseness is a concern, it is not as big of a concern as it might first seem. We did not focus on this point in this work because it is discussed in [22], and the way we bound the Lipschitz constant is not novel to this work. To summarize the points made in [22], while the spectral norm upper bound is found to be loose (compared to an empirical lower bound) by several orders of magnitude on normally-trained models, on GloRo Nets, the upper bound only exceeds empirical lower bounds by a factor of ~2 (and less, ~1.3x, in some cases). While this suggests there is room for improvement, as the certified radius could potentially be as much as doubled, the bound is tight enough that GloRo Nets are among the top state-of-the-art deterministic L2 certification methods. While perhaps surprising, these results can be explained by the fact that training is done directly against the method for taking the bound, encouraging networks for which the bound is sufficiently tight. We think there is good reason to believe that these results would carry over to RTK and Affinity GloRo Nets.
>
> > It is not clear to me why the proposed approaches could improve clean accuracy over the standard guarantee.
>
> We apologize for not making this more clear, but the clean accuracy means something slightly different on each of the models. Essentially, the clean accuracy is chosen to be match what one would measure if the \bot logit were ignored (put differently, assuming all points satisfy the respective robustness notion); and each model has a different notion of VRA according to the robustness definition used. For regular GloRo Nets, the “clean accuracy” means what it usually means---the number of points with the correct label. For RTK GloRo Nets, “clean accuracy” refers to the top-K accuracy (when the \bot logit is ignored); this is why it is consistently a lot higher for RTK models. For Affinity GloRo Nets, “clean accuracy” refers to a modification of top-k accuracy that counts a prediction as correct if all labels scored above the ground truth share a single affinity set with the ground truth. These metrics were briefly described in lines 246-249, but we will try to bring more attention to this in Table 1 to avoid confusion.
>
> > I am also concerned with the scalability of the certified models.
>
> While improving scalability is always a goal, we would argue that the GloRo-style approach is the most scalable deterministic certification method that we’re aware of. As demonstrated in [22], the additional time and memory cost (compared to standard training) of GloRo Nets is small, particularly compared to other methods,  and the overhead at test time is negligible. As discussed in our response to Reviewer 4, the modifications that we make to certify RTK and Affinity robustness do not introduce significant additional costs; we will collect data on this to report in follow-up during the discussion phase. Finally, Tiny-Imagenet is the largest application we’re aware of that has been evaluated on by a deterministic method for certification. Thus, we argue that RTK/Affinity GloRo Nets should have essentially the same ability to scale as standard GloRo Nets, which we believe have the best potential to scale among other deterministic certification methods.
>
> > Besides GloRo Nets, could the proposed properties be applied to other types of certified networks or certified training?
>
> This is certainly a possibility, and an interesting direction for future work that we will add to the discussion of this work. Below are a list of some different styles of methods and our thoughts on what might be possible. We would be happy to include a discussion of this sort in the camera-ready version.
>
> * **Randomized Smoothing:** Jia et al. suggest a technique based on randomized smoothing that's in the same spirit as our RTK robustness notion. While we point out several problems with their approach, Appendix A describes how to adapt their technique to get an equivalent property to our top-k robustness (Definition 2). However, we argue that RTK robustness (Definition 3) is the correct analogue to top-k accuracy. RTK robustness can be certified given a method to certify top-k accuracy, by looping over each k in {1, ..., K}. Thus, it should be possible to adapt the method of Jia et al. to correctly certify RTK robustness via randomized smoothing. A naive implementation of this would be very expensive, however, as it would require O(K^2) calls to Jia et al.'s method, which is already expensive. On hardware similar to our own, we estimate such an implementation would take about a minute to evaluate and certify each instance. Future work may be able to determine a more efficient way to achieve this. Note, too, that although RS can obtain certificates without any requirements on the original model, in order to get reasonable performance, it is necessary to train with Gaussian noise as part of the data augmentation. It is possible that additional training considerations would be important to make use of the relaxed guarantee in RS models.
> * **1-Lipschitz Models:** a similar certification approach to GloRo Nets uses orthonormal projections to ensure the model is 1-Lipschitz so that certification only requires a margin of epsilon between the top logit output and the others. Because these techniques achieve robustness through Lipschitzness, the certification aspect of the method described in Section 5 would also apply to such methods. Primarily these approaches differ from GloRo Nets in the way they are trained, targeting robustness though a hinge-like loss function that ensures a margin. Because of the nuances of RTK robustness, it is less clear how RTK robustness could be achieved with a simple loss function (since the network should have a choice in which k to target); however, the GloRo training we propose would still work to train models with orthonormalized kernels.
> * **Bound Propagation:** it is not immediately clear to us how to achieve RTK or Affinity robustness with methods like KW or IBP, but we believe this could be an interesting future direction. It may be possible to change these methods to ensure a margin between the kth class and the k+1th class. This in turn could be used as a step in certifying RTK robustness. Note however that this would likely mean that KW and IBP may require multiple propagations to handle RTK robustness, making them more expensive (while the analysis of the GloRo approach only needs to consider the logits, which only need to be computed once). An important issue, however,  is incorporating RTK robustness into the learning objective. Because of the disjunctive nature of RTK robustness, it is not obvious how to design the loss function to promote it. GloRo Nets avoid this issue because they essentially make an equivalence between the problem of certification and the robust learning objective.
>
> We note that while these are interesting directions, we chose GloRo Nets for this work because (1) they have recently been shown to be among the top state-of-the-art methods for deterministic certification, outperforming methods such as bound propagation; and (2) they admit an elegant way for incorporating our robustness notions into certifiable training, as certified training can be reduced to performing certification.

---

### Official Review · Reviewer_N8vP · 2021-07-17

**Rating:** 6
**Confidence:** 3

**Summary:**

The paper introduced two relaxed safety properties for classifiers: (1) relaxed top-k robustness, which is different from traditional top-1 classifier; and (2) affinity robustness, which specifies which sets of labels must be separated by a robustness margin, and which can be close a perturbation set. The experimental results on EuroSAT, CIFAR-100, Tiny-Imagenet and ACAS Xu demonstrated the effectiveness of the proposed method.

**Limitations And Societal Impact:**

No limitations and potential negative societal impact were found during my review.

**Main Review:**

Originality:

The proposed method basically based on the idea from global-robustness [22] but extended the method to a more general way, top-k accuracy or top set of labels robustness. The technique contribution is sound and straightforward, the idea also can be applied in some real-world use cases.
Specifically, the global-robustness is well defined in [22], the new problems proposed in this paper combining the idea from [12] but define a more general Top-k Robustness and Relaxed Top-K Robustness. It's an analog of changing the added class $\perp$ to the k-th class.
The implementation still using approximated upper bound of the global Lipschitz constant which is similar to [22].

Quality:

The paper is well-organized and easy to follow. Both theoretical analysis and experimental results are provided, especially the results on the EuroSAT dataset demonstrate the value of the Top-k Robustness.
After reading the paper I have some concerns about the experimental part:

1) The running time of the proposed method is not clear. Consider the paper claimed the efficiency in line 159,  the time complexity and robustness of the proposed method compared with existing top-1 global-robustness should be mentioned.
2) It looks like the $\epsilon$-ball is choose as $\ell_2$ norm perturbation, does the proposed method can be generalized to $\ell_\infty$ perturbation?
3) It's hard to evaluate the experimental results since all baselines are varieties of the GloRo Nets with fixed $\epsilon$. Also compare with some linear relaxation-based perturbation analysis methods like KW and IBP, they can also improve the top K robustness by constructing a specification matrix to improve the margin between a set of labels.

Significance:

To the best of my knowledge, the problem of certification in top-k robustness is novel and has real-world significance. But consider the method proposed in this work is based on existing work, global-robustness, with minor changes, so I doubt the novelty contribution on methodology.

During the rebuttal, I would like to raise my rating score if the authors can address my questions on 1) the implementation differences from [22] and 2) the running time of the proposed method comparing with nature training and global-robustness training.





**Time Spent Reviewing:**

4 hours

---

> ### Author Response · Authors · 2021-08-09
> **Thank you for your thoughtful review!**
>
> Thank you for your thoughtful review! We have replied to your specific comments below, and will aim to clarify these points in the final version of the text as well. We especially hope that our comments will sufficiently address your two main concerns, but would be happy to discuss further if necessary in the discussion phase.
>
> > [RTK robustness is] an analog of changing the added class to the k-th class.
>
> While this is a reasonable high-level description of what we do, we think it would be useful to point out that this characterization is not entirely accurate; and more specifically, it misses some of the key ideas that make our work a nontrivial extension of GloRo Nets [22]. Addressing this subtle point may also help clarify your questions regarding the differences from [22].
>
> To illustrate the point, let's say we are considering K = 2, and we have three logit outputs on a certain point where y1 > y2 > y3.
> Suppose that we define the \bot class based on y3, accounting for the Lipschitz constant, as suggested by the quote above.
> There are a couple ways to do this.
>
> * Suppose we want to ensure \bot doesn't pass y1. The problem here is similar to the problem we address with the method of Jia et al. [12]  (discussed in Appendix A). Basically, this would mean that y3 cannot pass y1, but y3 might be able to pass y2 still. Why is this a problem? To provide the analogue to top-2 accuracy, we would like to say that if the correct label is either 1 or 2 (in this case), the model was correct on this point. But to say that this is a *robust* correct prediction, we would like to know that if we make a small change to our point, the model will still be correct. If the correct label is 2, then this might not be the case here. So classes 1 and 2 need to be treated symmetrically, and we need to ensure that y3 cannot pass y1 *or* y2.
> * Suppose instead we want to ensure \bot doesn't pass y2, i.e., y3 can't pass y2. The first problem is that y3 may still be able to pass y1 (e.g., if y1 decreases rapidly, leaving us with the order y2' > y3' > y1' on the perturbed input). Even if we also account for this there is still one more issue. Namely, we want RTK robustness to be a relaxation of standard robustness, just like top-k accuracy is a relaxation of standard accuracy. Figure 1a in Section 3 shows that a model can be top-1 robust but not top-2 robust on some points. This is why our notion of RTK robustness instead checks if *any set of up to K classes* is robust, meaning that we also want to know if y2 can't pass y1 in case y3 *can* pass y2. This requires a bit more care than just slightly changing how \bot is defined in [22].
>
> Some of these subtleties about getting RTK robustness right are discussed in Section 3 and Appendix A, although we can add further discussion if it would help make the contributions of this work more clear; namely, to make it clearer that a sound analogue to top-k accuracy for robustness is actually fairly tricky to get right, as more straightforward approaches may be flawed.
>
> >  Implementation differences from [22]
>
> Some of the important subtleties that make our approach different from [22] are highlighted above. One of our key insights/contributions is showing that to get a true analogue to top-k accuracy, one must consider whether, for *any* k in {1, ... K}, *any class in the top k classes* can be passed by any class not in the top k classes (this important insight has been overlooked by prior work [12]). For each such k, the implementation is a fairly simple variation of [22], as GloRo Nets are a special case of this computation where k=1, but (1) we need to generalize the approach of [22] to other k besides 1 and (2) we need to consider all k in {1, ..., K}.
>
> Furthermore, implementing Affinity robustness in an efficient, vectorized way, required developing a masking algorithm to consider only pairs of classes that do not share an affinity set. While this was not presented in the paper (as we believe the algorithm is best described in code), we believe this implementation detail is nontrivial. If it would be helpful, we could include documented pseudo-code for this in an appendix (the documented code for the masking procedure is included in the supplementary material, and will be made available on a public repository when the paper is published).
>
> > Running time of the proposed method comparing with nature training and global-robustness training.
>
> In our experience training RTK/Affinity GloRo Nets, the runtime cost remains comparable to evaluating a normal model, as is also the case for GloRo Nets. We will run experiments to measure the exact cost on an apples-to-apples comparison of standard training, standard GloRo training, and RTK/Affinity GloRo training, and report the results. We are happy to provide this as follow-up during the discussion phase; please let us know if that would be helpful.
>
> In theory, the overhead compared to [22] comes from the fact that we must consider for each k in {1, …, K} whether any class in the top k classes can be passed by any class not in the top k classes. For each such k, the computation cost is essentially the same as that of GloRo Nets (since GloRo nets are essentially a special case of this computation where k=1), suggesting that the overhead would be a factor of K over the overhead of GloRo Nets, which is already reported to be quite small (approximately 5% increase over a standard model [22], but note that this cost does not scale with the size or depth of the model). Moreover, in our implementation, we use a vectorized approach that considers each of the K top classes in parallel, meaning that there is a memory overhead of K, but essentially no noticeable cost in computation time compared to GloRo Nets.
>
> > Does the proposed method can be generalized to linf perturbation?
>
> While our approach is designed to be particularly effective for the L2 norm, it can be easily generalized to the L_infinity norm as well. Leino et al. [22] discuss this in reference to GloRo Nets; essentially, the Lipschitz constant can be defined with respect to the linf norm, and upper-bounded. However, while the GloRo approach leads to better results than prior worth with L2, at the moment this does not appear to be true for L_infinity. This suggests that while GloRo-Net-style techniques are technically general to any Lp norm, more work is needed to determine their efficacy for cases other than L2. In this regard, the history of the prior literature perhaps suggests that L2 and linf may often be best solved using different techniques, as most of the best methods for certifying L_infinity robustness use techniques that don’t generalize to L2. We believe the L2 case is an important one, but considering RTK/Affinity robustness in the context of techniques optimized for linf is certainly an interesting direction.
>
> > Comparison to KW and IBP
>
> It is not immediately clear to us how to achieve RTK or Affinity robustness with these techniques, but we believe this could be an interesting future direction. We note that as mentioned, there are several important nuances with RTK robustness as opposed to top-k robustness (Definition 2) or the method proposed by Jia et al. [12]. Specifically, we don't only consider the margin between two sets of labels, but the margin between the top k labels and their complement *for all k in {1, ..., K}*. This means that KW and IBP may require multiple propagations to handle RTK robustness, making them more expensive (while the analysis of the GloRo approach only needs to consider the logits, which only need to be computed once).
>
> A more serious concern is incorporating RTK robustness into the learning objective of these techniques. Unless a model is trained against an RTK objective, we believe that it is unlikely that the RTK property will be provable, by any method. Because of the disjunctive nature of RTK robustness, it is not clear how to design the loss function to promote it. GloRo Nets avoid this issue because they essentially make an equivalence between the problem of certification and the robust learning objective, so that the disjunctive component is specified separately from the learning objective.
>
> We primarily focus on L2 certification. Kolter and Wong provide results for KW for L2, but [22] demonstrates that GloRo Nets are both more efficient and achieve notably higher VRA than KW. Leino et al. [22] also compared GloRo nets to BCP, which has commonalities with IBP (and succeeds it in the literature), and found similarly that GloRo Nets achieve better VRA with lower computational cost. However, we agree that modifying a technique like IBP to handle RTK or Affinity robustness would be interesting, and might lead to good L_infinity results.

---

> > ### Author Response · Authors · 2021-08-18
> > **Update**
> >
> > We were able to track running times for standard training, GloRo training, RT5 training, and Affinity training on CIFAR-100. The results are the following:
> >
> > | training method | time per epoch | time to evaluate test set |
> > |-----------------|----------------|---------------------------|
> > | standard        | 4.16s          | 0.55s                     |
> > | GloRo           | 5.68s          | 0.67s                     |
> > | RT5 GloRo       | 6.46s          | 0.67s                     |
> > | Affinity GloRo  | 6.66s          | 0.75s                     |
> >
> > As we can see, the overhead (in terms of time) of RTK over standard GloRo training is small during training (<15%), and negligible at test time. Similarly, Affinity adds only a small overhead over RTK (\~3% during training), though the overhead is a little bit higher at test time.
> >
> > We did not collect memory costs (this was unfortunately not straightforward with our version of tensorflow), but note that the memory overhead of GloRo over standard training was reported to be about 5% during training [22], which is notably smaller than the overhead in terms of time. Based on our analysis in our original response, we believe this suggests the memory overhead for RRT5 should not exceed 25% compared to standard training.
> >
> > We will update the camera-ready version with these numbers (along with the corresponding numbers for the other datasets).

---

> > > ### Comment · Reviewer_N8vP · 2021-08-26
> > > **Post rebuttal**
> > >
> > > Thanks for your response.
> > >
> > > I think most concerns are addressed in the rebuttal. I was a little bit overclaim on achieving global robustness via KW or IBP based linear relaxation based methods. The results convinced me well and I will raise my score to reflect them. Please including the clarification in the revison.

---

### Official Review · Reviewer_miwT · 2021-07-19

**Rating:** 6
**Confidence:** 4

**Summary:**

This paper proposes top-k robustness, where the top-k predictions should not change if the input is perturbed by a small amount. The authors propose an algorithm that extends GloRo to achieve certifiable (relaxed) top-k robustness.


**Limitations And Societal Impact:**

See above

**Main Review:**


Pro:

To my knowledge, the problem definition is novel and interesting. The authors propose that the top-k predictions should be robust to perturbations of the input. I really like the idea that for different samples, the model has certified robustness for different values of k, which seems to be a very useful flexibility when there is some label ambiguity.

The proposed algorithm to achieve the proposed top-k robustness seems very reasonable. Empirically the performance is better than the baselines.

Questions:

My main concern is the lack of discussion on the several problem parameters that are critical for understanding the performance of the algorithm. I believe these should be extensively discussed in the main paper.
1. There is very little mention of the maximum label set size K in the paper, which seems to be a critical hyper-parameter that affects performance. For example, if K is very large (e.g. close to the total number of classes) then the performance number is less meaningful than a small K.
2. The actual set size output by the model is also an important performance metric, which should be highlighted. A small prediction set is certainly preferable to a large prediction set
3. The choice of robustness norm ball size (denoted \epsilon in the paper) can be discussed.

There seems to be some ambiguity with the number of labels k to select. Traditionally the number of labels k is selected based on the predicted probability (e.g. select the smallest k such that the top-k labels account for 99% of the predicted probability). This paper proposes a different criteria for selecting k (some k where the top-k prediction set is robust to perturbations). This begs the question of which k to select in practice.

It would be interesting to see the connection to Lipschitness of the output logits. Suppose each logit vector f_i(x) is Lipschitz with respect to the input (e.g. this can be achieved by randomized smoothing), then would it be possible to naturally derive an RTK prediction function? (For example, just check that the top-k logit scores are far enough from the remaining logit scores). I would imagine that this will be a very competitive alternative.

Writing:

The writing is generally very clear and easy to follow. There are a few locations where clarity could be improved.
1. Section 4 needs more motivation. The jump from RTK to affinity robustness is somewhat abrupt.
2. The problem definition can be further clarified. Does the prediction model output an integer k along with a set of labels? In other words, should the prediction model know which set size k it is robust to. This is not clear from Definition 3 as the statement says “\exists k” rather than “there is an algorithm to find k.”
3. Following up on 2, Definition 3 should be written with more rigor. For example, is it \forall x \forall x’ \exists k, or \forall x’, \exists k, \forall x? These two have completely different interpretations. I do not think the current statement interpreted rigorously reflects the intuition the authors want to convey.

**Time Spent Reviewing:**

2 hours

---

> ### Author Response · Authors · 2021-08-09
> **Thank you for your thoughtful review!**
>
> Thank you for your thoughtful review! We have replied to your specific comments below, and will aim to clarify these points in the final version of the text as well. We will also try to make the writing more clear in the places you identify in your review.
>
> > The actual set size output by the model is also an important performance metric, which should be highlighted. A small prediction set is certainly preferable to a large prediction set.
>
> This is a good point; we will add this. One thing to note here is that it is possible for the network to output multiple robust set sizes (e.g., a point can be top-1 robust *and* top-2 robust). We will run some scripts to capture the number of points that admit each set size as a possible robust set size and report back with the results during the discussion phase.
>
> To get some intuition of what these results may look like: over the course of this work we had observed that when the *strictest* guarantee (i.e., the smallest set that is robust) is considered, over 75% of points can be assigned a top-1 guarantee on both CIFAR-100 and EuroSAT.
>
> It is fair to say that a smaller prediction set is better from the perspective of how robust a point is. There is also an argument to consider the *loosest* guarantee (i.e., the largest set that is robust), since this is the guarantee that will lead to the highest RTK VRA (having a larger robust set means it is more likely that one of the classes in the set is correct).
>
> > The choice of robustness norm ball size (denoted \epsilon in the paper) can be discussed.
>
> The value 0.141 is a standard robustness radius used frequently in the previous literature to evaluate l2 robustness on CIFAR-10; by extension, we use the same radius for CIFAR-100, which has the same input dimension. Although Tiny-Imagenet has slightly larger images than CIFAR (and might therefore deserve a larger radius), we note that in the prior literature that evaluates certifiable robustness on Tiny-Imagenet, 0.141 has also been used, so we thought it was the most appropriate radius to use for comparison. To our knowledge, EuroSAT has not been studied previously in the literature on robustness certification, but because its dimensionality is the same as that of Tiny-Imagenet, we kept the radius the same as well.
> For ACAS Xu in the appendix, the number of dimensions is far lower, and we were unaware of a standard radius to use. The radius we chose was based on an estimate of the minimum l2 distance between training points with different labels (essentially we assume that the training set can be perfectly robustly classified).
>
> > [How is K chosen?]
>
> We would argue that K is not truly a "hyperparameter," but is more similar to, e.g., the robustness radius (epsilon); that is, the selection of K is ideally not chosen based on performance considerations or trial and error, but rather on a consideration of *what safety requirements the given domain requires*. For example, if it would always be acceptable for the model to output as many as 5 classes that are interchangeable under small-norm perturbations, then 5 is an appropriate choice of K. The value for K thus depends on the domain but is ideally independent of its impact on the training outcome.
>
> > It would be interesting to see the connection to Lipschitness of the output logits. Suppose each logit vector f_i(x) is Lipschitz with respect to the input (e.g. this can be achieved by randomized smoothing), then would it be possible to naturally derive an RTK prediction function? (For example, just check that the top-k logit scores are far enough from the remaining logit scores). I would imagine that this will be a very competitive alternative.
>
> To clarify, we assume here that “Lipschitz” means 1-Lipschitz (i.e., the Lipschitz constant is 1). In this case there are approaches for certifiable training based on this, which have been found to be comparable in performance to GloRo Nets. These approaches typically project the kernels of the network to be orthonormal, which ensures 1-Lipschitzness. Because certification of such networks is also based on the Lipschitz constant, they could actually directly use our approach from Section 5, though this would make such an approach nearly identical to the one we propose. In essence, such methods are very similar to GloRo Nets, except they use a different loss and training procedure, and may impose different overheads during training and deployment.
>
> The idea of obtaining 1-Lipschitzness through RS is also interesting, and could work with our approach in a similar way, although it's worth pointing out that this would be particularly expensive to compute at runtime, as RS requires tens of thousands of samples to evaluate each instance.
>
> Consistent with this comment, and the comments of a few other reviewers, we will add a discussion of how RTK and Affinity robustness may be obtainable using other methods.
>
> > Does the prediction model output an integer k along with a set of labels?
>
> Our implementation does provide this output. Since the RTK GloRo Net is *certifiably* RTK robust, it should be viewed  as having an accompanying proof of RTK robustness for each prediction. This proof is constructive, and it specifies which k the network is RTK robust with at the given point.
>
> We agree that the ability to be robust with different k on different samples is an important and useful part of our approach; and it is interesting to think of including that in the definition of RTK robustness. The perspective we took, though, was that producing the value of k is a property of the certifiable network, rather than a requirement of RTK robustness. This is because we think of RTK robustness as a safety property, which a model may or may not have regardless of whether it can be proven. The job of the RTK GloRo Net is not only to achieve the property, but to prove that the property holds. As part of this proof it exhibits a satisfactory value of k.
>
> > Definition 3 should be written with more rigor. For example, is it \forall x \forall x’ \exists k, or \forall x’, \exists k, \forall x? These two have completely different interpretations.
>
> There is no universal quantifier over x. Perhaps this can be made more clear, but essentially we can think of Definition 3 as a property of some F, epsilon, and x; therefore x is fixed, rather than universally quantified.
>
> Ideally this property holds on as many x as possible (e.g., \forall x, Denition 3 holds), although depending on the choice of K and the decision boundary of F, some points may need to be rejected (i.e., some points may not satisfy Definition 3, though we would hope those points lie off the data manifold; e.g., see [this example](https://www.geogebra.org/geometry/guhx6dvt)).

---

> > ### Author Response · Authors · 2021-08-18
> > **Update**
> >
> > We were able to collect data on the guarantees achieved on each point, as requested. The results for EuroSAT and CIFAR-100 are provided below. In each case, we consider the fraction of certified points (i.e., non-rejected points) that received each guarantee. Note that the fractions do not sum to 1 because some points can receive multiple guarantees. For example, on EuroSAT, 56% of certified points received a guarantee for multiple values of $k$, and on CIFAR-100, 24% received a guarantee for multiple values of $k$.
> >
> > **EuroSAT:**
> >
> > | $k$                     |  1   |  2   |  3   |
> > |-------------------------|------|------|------|
> > | **fraction of points:** | 0.80 | 0.48 | 0.40 |
> >
> > **CIFAR-100:**
> >
> > | $k$                     |  1   |  2   |  3   |  4   |  5   |
> > |-------------------------|------|------|------|------|------|
> > | **fraction of points:** | 0.79 | 0.26 | 0.12 | 0.08 | 0.04 |
> >
> >
> > We see that the majority of points get a tight guarantee, with the looser guarantees being more rare with increasing looseness.
> >
> > We will include these results in the paper, probably plotted as a bar chart for better readability.

---

> > > ### Comment · Reviewer_miwT · 2021-09-03
> > > **Thank you**
> > >
> > > Thank you for the reply. This addressed my concerns.

---

### Official Review · Reviewer_veMk · 2021-07-24

**Rating:** 7
**Confidence:** 4

**Summary:**

This paper proposes two relaxations of local robustness: relaxed top-k robustness, and affinity robustness, which are more natural notions of robustness in certain settings. One asks for one of the top-k labels to be correct in a certain radius, and the other allows for certain similar classes to be perturbed to each other. This paper leverages prior work to train models that can be efficiently certified for each of the robustness properties, and shows new experimental results on common and relatively new datasets.

**Limitations And Societal Impact:**

Yes. I appreciate that this paper was one of the few papers I have seen that addresses ethics, even if it is just a few sentences.

**Main Review:**

Overall, this paper is borderline for me. I would rate it as a weak accept, and I would appreciate it if the authors could address some of my questions and concerns.

The paper is relatively original; although there are other works that try to establish similar notions of top-k robustness or hierarchical robustness (e.g., https://arxiv.org/abs/2102.09012, which the authors don’t cite, and https://arxiv.org/abs/1912.09899, which the authors do cite), the authors do so in a relatively new setting. The definitions that the authors propose are quite natural for certain problems, so I believe that the significance of the work is good, and the authors perform experiments on both very common datasets like CIFAR-100 and Tiny-ImageNet, as well as an interesting and relatively newer dataset called EuroSAT where their notions of robustness seem to work well. Overall, the paper is fairly clear.

I do have some questions and concerns, and would like to hear back from the authors regarding the following.
1) My main concern is that the certification method that the authors use is somewhat non-standard; they use GloRo Nets and a method that involves computing the Lipschitz constant between logits. While this leads to efficient certification and clean proofs, it is not clear how close to state of the art this method is, as the authors acknowledge such a method can lead to loose bounds in L198. Thus, I think it is important for the authors to add in a baseline and a comparison to more standard and state-of-the-art certification methods like using convex relaxations (https://arxiv.org/abs/1711.00851), or to explain why GloRo Nets are a better choice here. Next, a quick question; in L196 - is the approximation always an upper bound on the global Lipschitz constant, or is it sometimes less? It’s not clear to me from the writing if it’s always an upper bound but the bound is not tight, or if it’s an approximation of the value of an upper bound.
2) Table 1 - what does the “clean accuracy” column mean for the “CIFAR-100 standard” and “CIFAR-100 RT5” rows? Does each row correspond to a model trained in a different way, but evaluated in the same way when measuring “clean accuracy”? If so, how is the “clean accuracy” so low for “CIFAR-100 standard”? I would expect that standard training should have the highest (or at least very close to the highest) clean accuracy.
3) Figure 3 - please add a vertical line separating the images in (b) and the images in (c)
4) L134-136: The paper discusses something that has not been presented clearly yet (the hierarchical nature of the resulting RTK-trained model, which the authors discuss more in section 6.3). Perhaps the authors could include a reference to the part of the text that presents this result more clearly?
5) Section 6.3 - what does the hierarchy have to do with the RT5 training? Does the same hierarchy occur even for a model trained with a standard loss?


**Time Spent Reviewing:**

3

---

> ### Author Response · Authors · 2021-08-09
> **Thank you for your thoughtful review!**
>
> Thank you for your thoughtful review! We have replied to your specific comments below, and will aim to clarify these points in the final version of the text as well.
>
> > I think it is important for the authors to add in a baseline and a comparison to more standard and state-of-the-art certification methods like using convex relaxations (https://arxiv.org/abs/1711.00851), or to explain why GloRo Nets are a better choice here.
>
> Our primary focus in this work was (deterministic) certification of L2 robustness. For this norm, GloRo Nets are actually more performant than KW (the method cited above). E.g., [22] reports that GloRo Nets achieve a VRA that is 8 percentage points above KW on CIFAR-10 and 18 percentage points above KW on MNIST (using standard values of epsilon that are used commonly in the literature). Moreover, [22] is a fairly recent publication with comparisons to many prior approaches, and performs better or comparably to all other deterministic methods surveyed in the paper. We therefore think that the GloRo approach is a good representation of state-of-the-art L2 certifiably-robust methods.
>
> > While this leads to efficient certification and clean proofs, it is not clear how close to state of the art this method is, as the authors acknowledge such a method can lead to loose bounds in L198.
>
> While potential looseness is a concern, it is not as big of a concern as it might first seem. We did not focus on this point in this work because it is discussed in [22], and the way we bound the Lipschitz constant is not novel to this work. To summarize the points made in [22], while the spectral norm upper bound is found to be loose (compared to an empirical lower bound) by several orders of magnitude on normally-trained models, on GloRo Nets the upper bound only exceeds the empirical lower bounds by a factor of ~2 (and less, ~1.3x, in some cases). While this suggests there is room for improvement, as the certified radius could potentially be as much as doubled, the bound is tight enough that GloRo Nets are among the top state-of-the-art deterministic L2 certification methods. While perhaps surprising, these results can be explained by the fact that training is done directly against the method for taking the bound, encouraging networks for which the bound is sufficiently tight. We think there is good reason to believe that these results would carry over to RTK and Affinity GloRo Nets, but we would be able to provide a similar comparison of upper and lower bounds on our models if it would be helpful.
>
> > In L196 - is the approximation always an upper bound on the global Lipschitz constant, or is it sometimes less?
>
> It’s always an upper bound, provided that the power method (used to compute the spectral norm) has converged. During training we don’t necessarily run to convergence (as this will need to be done after each update), but immediately after training it is run to convergence (to a precision of $10^{-4}$; while the bound obtained is typically of the order of $10^0$-$10^1$), since this only needs to be done once. Thus the VRAs reported can correctly be understood as a true guarantee.
>
> > Table 1 - what does the “clean accuracy” column mean for the “CIFAR-100 standard” and “CIFAR-100 RT5” rows?
>
> We apologize for not making this more clear, but the clean accuracy means something slightly different on each of the models. Essentially, the clean accuracy is chosen to be match what one would measure if the \bot logit were ignored (put differently, assuming all points satisfy the respective robustness notion); and each model has a different notion of VRA according to the robustness definition used. For regular GloRo Nets, the “clean accuracy” means what it usually means---the number of points with the correct label. For RTK GloRo Nets, “clean accuracy” refers to the top-K accuracy (when the \bot logit is ignored); this is why it is consistently a lot higher for RTK models. For Affinity GloRo Nets, “clean accuracy” refers to a modification of top-k accuracy that counts a prediction as correct if all labels scored above the ground truth share a single affinity set with the ground truth. These metrics were briefly described in lines 246-249, but we will try to bring more attention to this in Table 1 to avoid confusion.
>
> > Section 6.3 - what does the hierarchy have to do with the RT5 training? Does the same hierarchy occur even for a model trained with a standard loss?
>
> This is a good point. We will run experiments to answer this question, and report back during the discussion phase; however, also note that the “groups” formed in RTK/Affinity models are certifiably robust, while it is unlikely that the same could be said of standard training or even standard certifiable training. We agree, though, that it is probably fair to say that classes sharing an affinity set are more likely to have similarly-ranked logit scores even in standard-trained models.

---

> > ### Comment · Reviewer_veMk · 2021-08-26
> > **Thank you for your response**
> >
> > I appreciate the clarifications - I hope that you are able to include them in the final version of the paper.
> >
> > I did not realize that GloRo Nets were close to the state-of-the-art for deterministic L2-robustness certification, as most L2-robustness certification work is randomized. The author's choice to use them now makes more sense to me.
> >
> > The clarification about "clean accuracy" appears to have confused more than just myself, so I would highly recommend the authors keep that clarification in any updated versions.
> >
> > I have increased my score slightly to reflect this.

---

### Official Review · Reviewer_TzLt · 2021-07-24

**Rating:** 6
**Confidence:** 3

**Summary:**

This work proposes two properties for certified robustness (local) -- relaxed top-k robustness, and affinity robustness. The importance of such properties is discussed in light of how many datasets contain some class hierarchy (such as superclasses), or inputs that may belong to multiple classes. The introduction of such relaxations is proposed to serve as a means for being analogs to the top-k robustness or directed attack robustness, which are often discussed in empirical settings. The paper also demonstrates how these objectives can be put to practice by making modifications to the training objective of GloRo Nets. These relaxed robust networks are also shown to demonstrate properties of class hierarchy.



**Limitations And Societal Impact:**

Yes

**Main Review:**

### Writing
The paper is well written and easy to follow. The extensive case-studies of datasets are helpful in motivating the method to the reader. However, I believe that this comes at the expense of *significant* technical detail. In a revised version, I would expect to see a much larger amount of details regarding how the method can be re-implemented (Section 5.2).

### Significance
This work would be useful to the progress of certifiable robustness to larger datasets, especially those that contain class similarities and hierarchies.

### Problems
1. Considering that CIFAR100 superclass (20 classes) is a significantly relaxed version of the CIFAR100 problem, I would have expected the certified accuracies to be closer to CIFAR10. However, the reported numbers appear much lower? This makes it difficult to quantify the utility of the relaxation. Can you provide a comparison for CIFAR10 and CIFAR100 (say 10 super classes of CIFAR100) on the same norm and same radius? Is the lack of certified accuracy because of resorting to GloRo nets?
2. Can this also be adapted to non-l2 norms? Moreover, I could not find anywhere in the paper where the authors clearly specified the norm type (apart from using the symbol, which is often used to denote any lp norm in literature).


### Questions
1. The paper highlights how "the classes that are “grouped” by the model typically follow a logical structure, even without supervision". What is your observation of this logical structure in (a) standard training; and (b) standard locally robust training. It is fair to say that this is a phenomenon that occurs *exclusively* because of RTK training? I would expect otherwise.

2. Do you think there may exist adaptations of other certification techniques (apart from GloRo nets) that can incorporate these objectives in future work? This would include both deterministic and stochastic certification paradigms.

**Time Spent Reviewing:**

6

---

> ### Author Response · Authors · 2021-08-09
> **Thank you for your thoughtful review!**
>
> Thank you for your thoughtful review! We have replied to your specific comments below, and will aim to clarify these points in the final version of the text as well.
>
> > The extensive case-studies of datasets are helpful in motivating the method to the reader. However, I believe that this comes at the expense of significant technical detail. In a revised version, I would expect to see a much larger amount of details regarding how the method can be re-implemented (Section 5.2).
>
> Thanks for this feedback; we will add more technical detail about the implementation to Section 5 in the camera-ready version (and we will also be making our code public for reproducibility).
>
> > Considering that CIFAR100 superclass (20 classes) is a significantly relaxed version of the CIFAR100 problem, I would have expected the certified accuracies to be closer to CIFAR10. Can you provide a comparison for CIFAR10 and CIFAR100 (say 10 super classes of CIFAR100) on the same norm and same radius?
>
> Note that VRA with RTK robustness is still harder than classifying with 1/K classes. This is because the loosest RTK guarantee on a given point might be k < K, in which case the correct class for that instance would need to be among the top k classes rather than the top K. E.g., some points receive only a top-1 guarantee even under the relaxed variant, meaning that for those points to count towards the VRA, they would need to be classified exactly correctly, not simply in the same superclass.
> For the sake of this comparison, we will collect results for RT10 robustness on CIFAR-100, and we will also collect data on the breakdown of guarantees that are achieved (as was also requested by Reviewer 3).
>
> > Can this also be adapted to non-l2 norms?
>
> Both the definitions of RTK and Affinity robustness, as well as our approach for certifiable training are indeed general beyond the L2 norm, which is why sections 3 and 4 use a generic Lp norm notation. However, our focus in this work, particularly on the empirical side, was L2 robustness certification, and we will try to make this more clear, especially in the evaluation section.
>
> Leino et al. [22] discuss L_infinity certification via standard GloRo Nets (this discussion also applies to our approach); essentially, the Lipschitz constant can be defined with respect to the L_infinity norm, and upper-bounded. However, while the GloRo approach leads to better results than prior worth with L2, at the moment this does not appear to be true for L_infinity. This suggests that while GloRo-Net-style techniques are technically general to any Lp norm, more work is needed to determine their efficacy for cases other than L2. In this regard, the history of the prior literature perhaps suggests that L2 and L_infinity may often be best solved using different techniques, as most of the best methods for certifying L_infinity robustness use techniques that don’t generalize to L2. We believe the L2 case is an important one, but considering RTK/Affinity robustness in the context of techniques optimized for linf is certainly an interesting direction.
>
> > The paper highlights how “the classes that are “grouped” by the model typically follow a logical structure, even without supervision”. What is your observation of this logical structure in (a) standard training; and (b) standard locally robust training?
>
> This is a good question. We will run experiments to answer this question; however, also note that the "groups" formed in RTK/Affinity models are *certifiably robust*, while it is unlikely that the same could be said of standard training or even standard certifiable training. We agree, though, that it is probably fair to say that classes sharing an affinity set are more likely to have similarly-ranked logit scores even in standard-trained models.
>
> > Do you think there may exist adaptations of other certification techniques (apart from GloRo nets) that can incorporate these objectives in future work?
>
> This is certainly a possibility, and an interesting direction for future work that we will add to the discussion of this work. Below are a list of some different styles of methods and our thoughts on what might be possible. We would be happy to include a discussion of this sort in the camera-ready version.
>
> * **Randomized Smoothing:** Jia et al. suggest a technique based on randomized smoothing that's in the same spirit as our RTK robustness notion. While we point out several problems with their approach, Appendix A describes how to adapt their technique to get an equivalent property to our top-k robustness (Definition 2). However, we argue that RTK robustness (Definition 3) is the correct analogue to top-k accuracy. RTK robustness can be certified given a method to certify top-k accuracy, by looping over each k in {1, ..., K}. Thus, it should be possible to adapt the method of Jia et al. to correctly certify RTK robustness via randomized smoothing. A naive implementation of this would be very expensive, however, as it would require O(K^2) calls to Jia et al.'s method, which is already expensive. On hardware similar to our own, we estimate such an implementation would take about a minute to evaluate and certify each instance. Future work may be able to determine a more efficient way to achieve this. Note, too, that although RS can obtain certificates without any requirements on the original model, in order to get reasonable performance, it is necessary to train with Gaussian noise as part of the data augmentation. It is possible that additional training considerations would be important to make use of the relaxed guarantee in RS models.
> * **1-Lipschitz Models:** a similar certification approach to GloRo Nets uses orthonormal projections to ensure the model is 1-Lipschitz so that certification only requires a margin of epsilon between the top logit output and the others. Because these techniques achieve robustness through Lipschitzness, the certification aspect of the method described in Section 5 would also apply to such methods. Primarily these approaches differ from GloRo Nets in the way they are trained, targeting robustness though a hinge-like loss function that ensures a margin. Because of the nuances of RTK robustness, it is less clear how RTK robustness could be achieved with a simple loss function (since the network should have a choice in which k to target); however, the GloRo training we propose would still work to train models with orthonormalized kernels.
> * **Convex Relaxation/Bound Propagation:** it is not immediately clear to us how to achieve RTK or Affinity robustness with methods like KW or IBP, but we believe this could be an interesting future direction. It may be possible to change these methods to ensure a margin between the kth class and the k+1th class. This in turn could be used as a step in certifying RTK robustness. Note however that this would likely mean that KW and IBP may require multiple propagations to handle RTK robustness, making them more expensive (while the analysis of the GloRo approach only needs to consider the logits, which only need to be computed once). An important issue, however,  is incorporating RTK robustness into the learning objective. Because of the disjunctive nature of RTK robustness, it is not obvious how to design the loss function to promote it. GloRo Nets avoid this issue because they essentially make an equivalence between the problem of certification and the robust learning objective.
>
> We note that while these are interesting directions, we chose GloRo Nets for this work because (1) they have been shown to be among the top state-of-the-art methods for deterministic certification, matching or outperforming the l2 VRA of all other deterministic methods recently surveyed by Leino et al.; and (2) they admit an elegant way for incorporating our robustness notions into certifiable training, as certified training can be reduced to performing certification.

---

> > ### Comment · Reviewer_TzLt · 2021-08-30
> > **Follow up**
> >
> > Thank you for the detailed response. I look forward to seeing the CIFAR10 vs CIFAR100 10 super class comparison in the final version

---

### Decision · Program_Chairs · 2021-09-27

**Decision:**

Accept (Poster)

**Comment:**

The authors propose relaxations of robustness which deal with the fact that tasks can have varying label ambiguity. All reviewers agreed on the quality of the work and its relevance to security/safety critical settings. In particular, this definition will likely help scale certification approaches to larger datasets where label ambiguity is more common. In addition to editing the paper to include new discussion presented in their responses, I encourage the authors to make their code public to facilitate reproducibility. In particular, it will be important to clarify in the writing that the focus in this paper is on deterministic certification of L2 robustness so that the choice of GloRo Nets is better motivated. The authors are also encouraged to outline clearly how follow-up on the work may study other certification techniques as suggested by the discussion with reviewer TzLt.